Hydrology and Earth System Sciences
EGU
Discussions

# Technical Note – RAT: a Robustness Assessment Test for calibrated and uncalibrated hydrological models

Pierre Nicolle[1,3], Vazken Andréassian[1,*], Paul Royer-Gaspard[1], Charles Perrin[1], Guillaume Thirel[1], Laurent Coron[2], Léonard Santos[1]

[1]Université Paris-Saclay, INRAE, UR HYCAR, 92160, Antony, France

[2]EDF, DTG, Toulouse, France

[3]now at Université Gustave Eiffel, Nantes, France

[*]Corresponding author: Vazken Andréassian (vazken.andreassian@inrae.fr)

## Key Words
hydrological modelling, split-sample test, differential split-sample test, model evaluation, robustness, climate change

## Key Points

- a new method (RAT) is proposed to assess the robustness of hydrological models, as an alternative to the classical split-sample test
- the RAT method does not require multiple calibrations: it is therefore applicable to uncalibrated models
- the RAT method can be used to determine whether a hydrological model can be safely used for climate change impact studies
- success at the RAT test is a necessary (but not sufficient) condition of model robustness

## Abstract
Prior to their use under future changing climate conditions, all hydrological models should be thoroughly evaluated regarding their temporal transferability (application in different time periods) and extrapolation capacity (application beyond the range of known past conditions). This note presents a straightforward evaluation framework aimed at detecting potential undesirable climate dependencies in hydrological models: the robustness assessment test (RAT). Although it is conceptually inspired by the classic differential split-sample test of Klemeš (1986), the RAT presents the advantage to be applicable to all types of models, be they calibrated or not (i.e. regionalized or physically based). In this note, we present the RAT, illustrate its application on a set of 21 catchments, verify its applicability hypotheses and compare it to previously published tests. Results show that the RAT is an efficient evaluation approach, passing it successfully can be considered a prerequisite for any hydrological model to be used for climate change impact studies.

Hydrology and Earth System Sciences
EGU
Discussions

# 1 Introduction

## 1.1 All hydrological models should be evaluated for their robustness

Hydrologists are increasingly requested to provide predictions of the impact of climate change (Wilby, 2019). Given the expected evolution of climate conditions, the actual ability of models to predict the corresponding evolution of hydrological variables should be verified (Beven, 2016). Indeed, when using a hydrological model for climate change impact assessment, we make two implicit hypotheses concerning:

- the **capacity of extrapolation beyond known hydroclimatic conditions**: we assume that the hydrological model used is able to extrapolate catchment behaviour under conditions not or rarely seen in the past. While we do not expect hydrological models to be able to simulate a behaviour which would result from a modification of catchment physical characteristics, we do expect them to be able to represent the catchment response to extreme climatic conditions (and possibly to conditions more extreme than those observed in the past);

- the **independence of the model set-up period**: we assume that the model functioning is independent of the climate it experienced during its set-up/calibration period. For those models which are calibrated, we assume that the parameters are generic and not specific to the calibration period, i.e. they do not suffer from overcalibration on this period (Andréassian et al., 2012).

Hydrologists make the hypothesis that model structure and parameters are well-identified over the calibration period and that parameters remain relevant over the future period, when climate conditions will be different. Unfortunately, the majority of hydrological models are not entirely independent of climate conditions (Refsgaard et al., 2013; Thirel et al., 2015). When run under changing climate conditions, they sometimes reveal an unwanted sensitivity to the data used to conceive or calibrate them (Coron et al., 2011).

The diagnostic tool most widely used to assess the robustness of hydrological models is the split-sample test (SST) (Klemeš, 1986), which is considered by all hydrologists as a "good modelling practice" (Refsgaard & Henriksen, 2004). The SST stipulates that when a model requires calibration (i.e. when its parameters cannot be deduced directly from physical measurements or catchment descriptors), it should be evaluated twice: once on the data used for calibration and once on an independent dataset. This practice has been promoted in hydrology by Klemeš (1986), who did not invent the concept (Arlot & Celisse, 2010; Larson, 1931; Mosteller & Tukey, 1968), but who formalized it for hydrological modelling. Klemeš proposed initially a four-level testing scheme for evaluating model transposability in time and space: (i) split-sample test on two independent periods, (ii) proxy-basin test on neighbouring catchments, (iii) differential split-sample test on contrasted independent periods (DSST), and (iv) proxy-basin differential split-sample test on neighbouring catchments and contrasted periods.

For model applications in a changing climate context, Klemeš's DSST procedure is of particular interest. Indeed, when calibration and evaluation are done over climatically-contrasted past periods, the model faces the difficulties it will have to deal with in the future. The power of DSST can be limited by the climatic variability observed in the past, which may be far below the drastic changes

expected in the future. However, a satisfactory behaviour during the DSST can be seen as a prerequisite of model robustness.

### 1.2 Past applications of the DSST method

The DSST received limited attention up to the 2010s, with only a few studies which applied it. The studies by Refsgaard & Knudsen (1996) and Donelly-Makowecki & Moore (1999) investigated to which extent Klemeš's hierarchical testing scheme could be used to improve the conclusions of model intercomparisons. Though the authors of the first study did not find large differences between the SST and DSST when comparing conceptual and physically-oriented models, the authors of the second study found that the DSST was more powerful than the SST to discriminate between four event-based models. The study by Xu (1999) questioned the applicability of models in nonstationary conditions and was one of the early attempts to apply the Klemeš's testing scheme in this perspective. Similarly, tests carried out by Seibert (2003) explicitly intended to test the ability of a model to extrapolate beyond calibration range and showed limitations of the tested model, stressing the need for improved calibration strategies. Last, Vaze et al. (2010) also investigated the behaviour of four rainfall-runoff models under contrasting conditions, using wet and dry periods on catchments in Australia that experienced a prolonged drought period. They observed different model behaviours when going from wet to dry or dry to wet conditions.

More recently, Coron et al. (2012) proposed a generalized SST (GSST) allowing for an exhaustive DSST to evaluate model transposability over time under various climate conditions. The concept of GSST consists in testing "the model in as many and as varied climatic configurations as possible, including similar and contrasted conditions between calibration and validation. […] The GSST procedure simply consists of a series of calibration-validation tests on subperiods of equal length, considering all possible configurations". Seifert et al. (2012) used a differential split-sample approach to test a hydrogeological model (differential being understood with respect to differences in groundwater abstractions). Li et al. (2012) identified two dry and two wet periods in long hydroclimatic series to understand how a model should be parameterised to work under nonstationary climatic conditions. Teutschbein and Seibert (2013) performed differential split-sample tests by dividing the data series into cold and warm as well as dry and wet years, in order to evaluate bias correction methods. Thirel et al. (2015) put forward an SST-based protocol to investigate how hydrological models deal with changing conditions, which was widely used during an IAHS workshop, both with physically-oriented models (Gelfan et al., 2015; Magand et al, 2015), conceptual models (Brigode et al., 2015; Efstratiadis et al., 2015; Hughes, 2015; Kling et al., 2015; Li et al., 2015; Yu and Zhu, 2015) or data-based models (Tanaka and Tachikawa, 2015; Taver et al., 2015).

Recently, with the growing concern on model robustness in link with the Panta Rhei decade of the International Association of Hydrological Sciences (IAHS) (Montanari et al., 2013), a slow but steadily increasing interest is noticeable for procedures inspired by Klemeš's DSST (see e.g. the Unsolved Hydrological Problem n° 19 in the paper by Blöschl et al., 2019: *How can hydrological models be adapted to be able to extrapolate to changing conditions?*). A few studies used the original DSST or GSST to implement more demanding model tests (Bisselink et al., 2016; Gelfan and Millionshchikova, 2018; Rau et al., 2019; Vormoor et al., 2018). For example, based on an ensemble approach using six hydrological models, Broderick et al. (2016) investigated under DSST conditions how the robustness

can be improved by multi-model combinations. They recommend selecting the best available analogues of expected annual mean and seasonal climate conditions.

A few authors also tried to propose improved implementations of these testing schemes. Seiller et al. (2012) used non-continuous periods or years selected on mean temperature and precipitation to enhance the contrast between testing periods. This idea to jointly use these two climate variables to select periods was further investigated by Gaborit et al. (2015), who assessed how the temporal model robustness can be improved by advanced calibration schemes. They showed that the robustness of the tested model was improved when going from humid-cold to dry-warm or from dry-cold to humid-warm conditions when using regional calibration instead of local calibration. Dakhlaoui et al. (2017) investigated the impact of DSST on model robustness by selecting dry/wet and cold/hot hydrological years to increase the contrast in climate conditions between calibration and validation periods. These authors later proposed a bootstrap technique to widen the testing conditions (Dakhlaoui et al. 2019). The investigations of Fowler et al. (2018) identified some limits of the DSST procedure and concluded that “model evaluation based solely on the DSST is hampered due to contingency on the chosen calibration method, and it is difficult to distinguish which cases of DSST failure are truly caused by model structural inadequacy”. Last, Motavita et al. (2019) combined DSST with periods of variable length, and conclude that parameters obtained on dry periods may be more robust.

All these past studies show that there is still methodological work needed on the issue of model testing and robustness assessment. This note is a further step in that direction.

### 1.3 Scope of the technical note

This note presents a new generic diagnostic framework inspired by Klemeš’s DSST procedure and by our own previous attempts (Coron et al., 2012; Thirel et al., 2015a) to assess whether a hydrological model can be considered “climate-proof”. One of the problems of existing methods is the requirement of multiple calibrations: these are relatively easy to implement with parsimonious conceptual models but definitively not with complex models that require long interventions by expert modellers and, obviously, not for those models with a once-for-all parameterisation.

Here, we propose a framework that is applicable with only one long period for which a model simulation is available. Thus, the proposed test is even applicable to those models that do not require calibration (or to those for which only a single calibration exists).

Section 2 presents and discusses the concept of the proposed test, section 3 presents the catchment set and the evaluation method, and section 4 illustrates the application of the test on a set of French catchments, with a comparison to a reference procedure.

## 2 The robustness assessment test (RAT) concept

The robustness assessment test (RAT) proposed in this note is inspired by the work of Coron et al. (2014). The specificity of the RAT is that it requires only one calibration (or one parameterisation) covering a sufficiently-long period (at least 30 years) with as much climatic variability as possible. Thus, it applies at the same time to simple conceptual models that can be calibrated automatically, to more complex models requiring expert calibration, and to uncalibrated models for which

parameters are derived from the measurement of certain physical properties. The RAT consists in computing a relevant numeric criterion repeatedly each year and then exploring its correlation with a climatic factor deemed meaningful, in order to identify undesirable dependencies and thus to assess the extrapolation capacity (Roberts et al., 2017) of any hydrological model. Indeed, if the performances of a model are shown to be dependent on a given climate variable, this can be an issue when the model is used on a period with a changing climate. The flowchart in Figure 1 summarizes the concept.

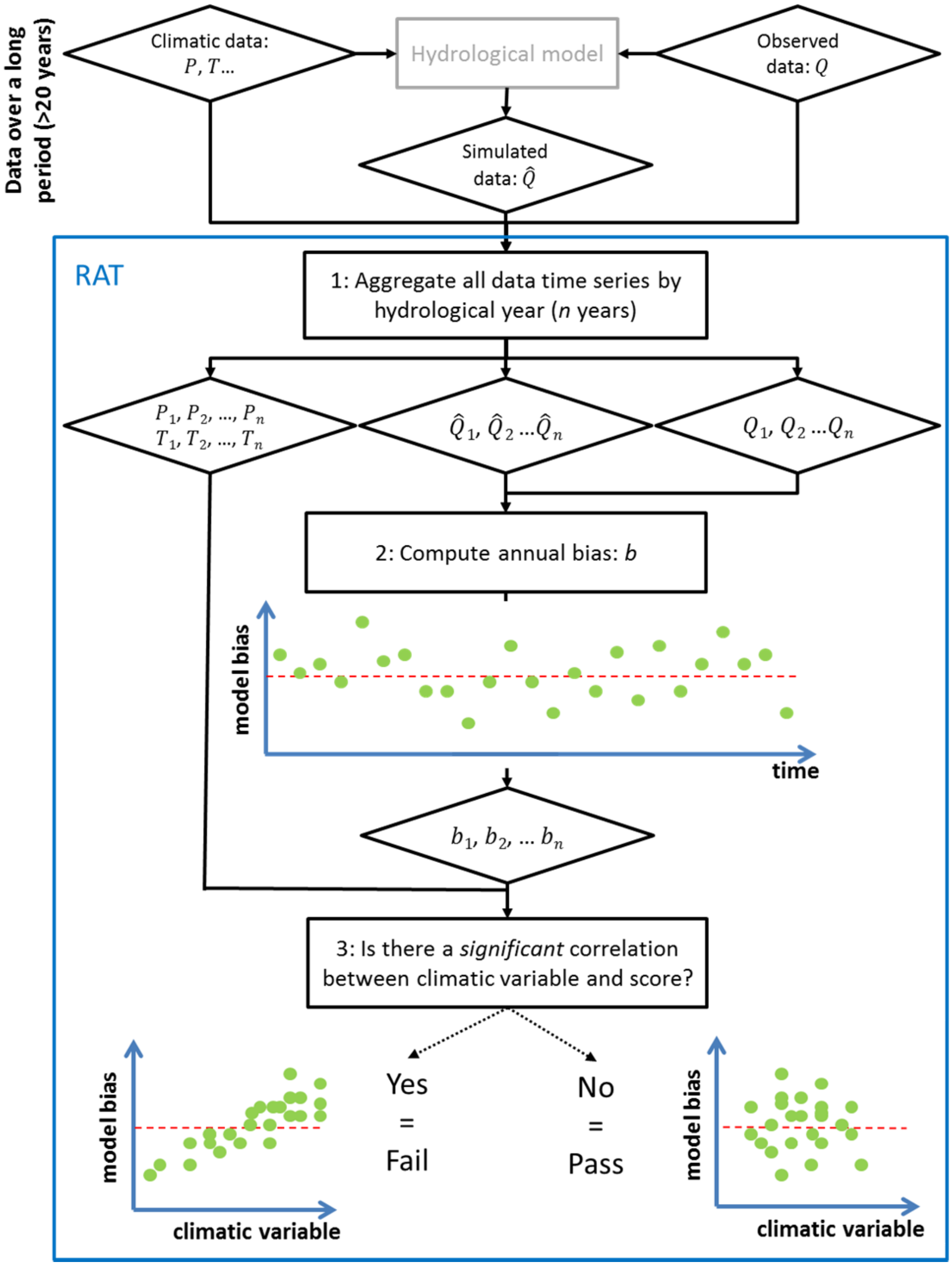

159

160 **Figure 1. Flow chart of the Robustness Assessment Test**

161 An example is shown in Figure 2, with a daily time step hydrological model calibrated on a 47-year
162 streamflow record. Note that this plot could be obtained from any hydrological model calibrated or
163 not. The relative streamflow bias ($(\overline{Q_{sim}}/\overline{Q_{obs}} - 1)$, with $\overline{Q_{sim}}$ and $\overline{Q_{obs}}$ being the mean simulated
164 and observed streamflows respectively) is calculated on an annual basis (47 values in total). Then,
165 the annual bias values are plotted against climate descriptors, typically the annual temperature

absolute anomaly ($T - \overline{T}$, where $T$ is the annual mean and $\overline{T}$ is the long-term mean annual temperature), the annual precipitation relative anomaly $P/\overline{P} - 1$ and the humidity index relative anomaly $HI/\overline{HI} - 1$, where $HI = P/E_0$, $E_0$ being the potential evaporation). Note that the mean annual values are computed on hydrological years (here from August 1st of year *n-1* to July 31st of year *n*). In this example, there is a slight dependency of model bias on precipitation and humidity index. Clearly, this could be a problem if we were to use this model in an extrapolation mode.

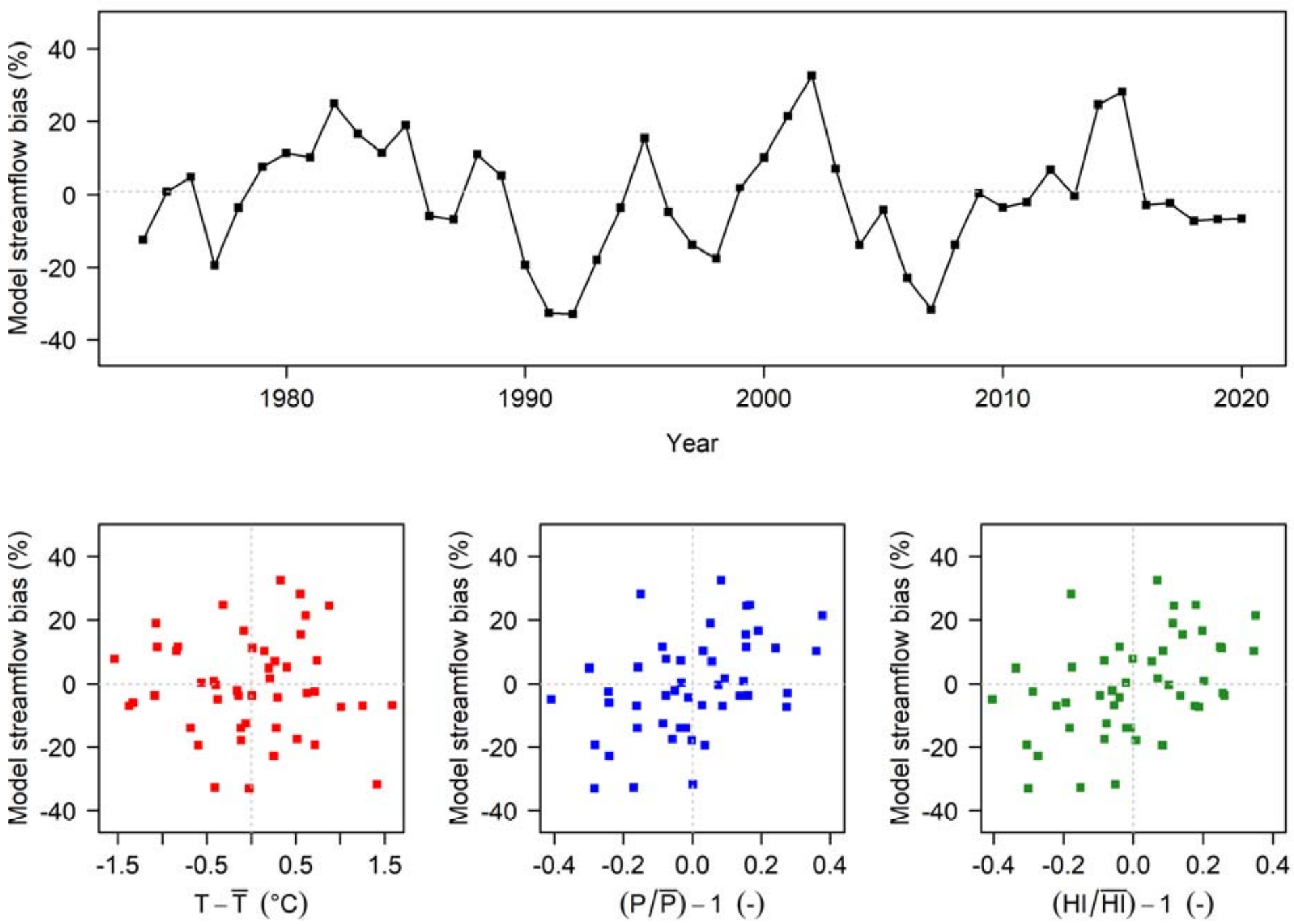

**Figure 2. Robustness Assessment Test (RAT) applied to a hydrological model: the upper graph presents the evolution in time (year by year) of model streamflow bias; the lower scatterplots present the relationship between model bias and climatic variables (temperature *T*, precipitation *P* and humidity index *HI*, from left to right)**

Whereas the methods based on the split-sample test (i.e. Coron et al, 2012 and Thirel et al., 2015b) evaluate model robustness on periods that are independent of the calibration period, it is not the case for the RAT. Consequently, one could fear that the results of the RAT evaluation may be influenced by the calibration process. However, because the RAT uses a very long period for calibration, we hypothesize that the weight of each individual year in the overall calibration process is small, almost negligible. This assumption can be checked by comparing the RAT with a leave-one-out SST (see Appendix). The analysis showed that this hypothesis is reasonable for long time series, but that the RAT is not applicable when the available time period is too short (less than 20 years).

Last, we would like to mention that the RAT procedure is different from the Proxy metric for Model Robustness (PMR) presented by Royer-Gaspard et al. (2021), even if both methods aim to evaluate hydrological model robustness without employing a multiple calibrations process: the PMR is a simple metric to estimate the robustness of a hydrological model, while the RAT is a method to

diagnose the dependencies of model errors to certain types of climatic changes. Thus, the RAT and the PMR may be seen as complementary tools to assess a variety of aspects about model robustness.

# 3 Material and methods

## 3.1 Catchment set

We employed the dataset previously used by Nicolle et al. (2014), comprising 21 French catchments (Figure 3), with complementary data until 2020. Catchments were chosen to represent a large range of physical and climatic conditions in France, with sufficiently-long observation time series (daily streamflow from 1974 to 2020) in order to provide a diverse representation of past hydroclimatic conditions. Streamflow data come from the French HYDRO database (Leleu et al., 2014) and with quality control performed by the operational hydrometric services. Catchment size ranges from 380 to 4,300 km² and median elevation from 70 to 1020 m.

The daily precipitation and temperature data originate from the gridded SAFRAN climate reanalysis (Vidal et al., 2010) over the 1959–2020 period. More information about the catchment set can be found in Nicolle et al. (2014). Aggregated catchment files and computation of Oudin potential evaporation (Oudin et al., 2005) was made as described in Delaigue et al. (2018).

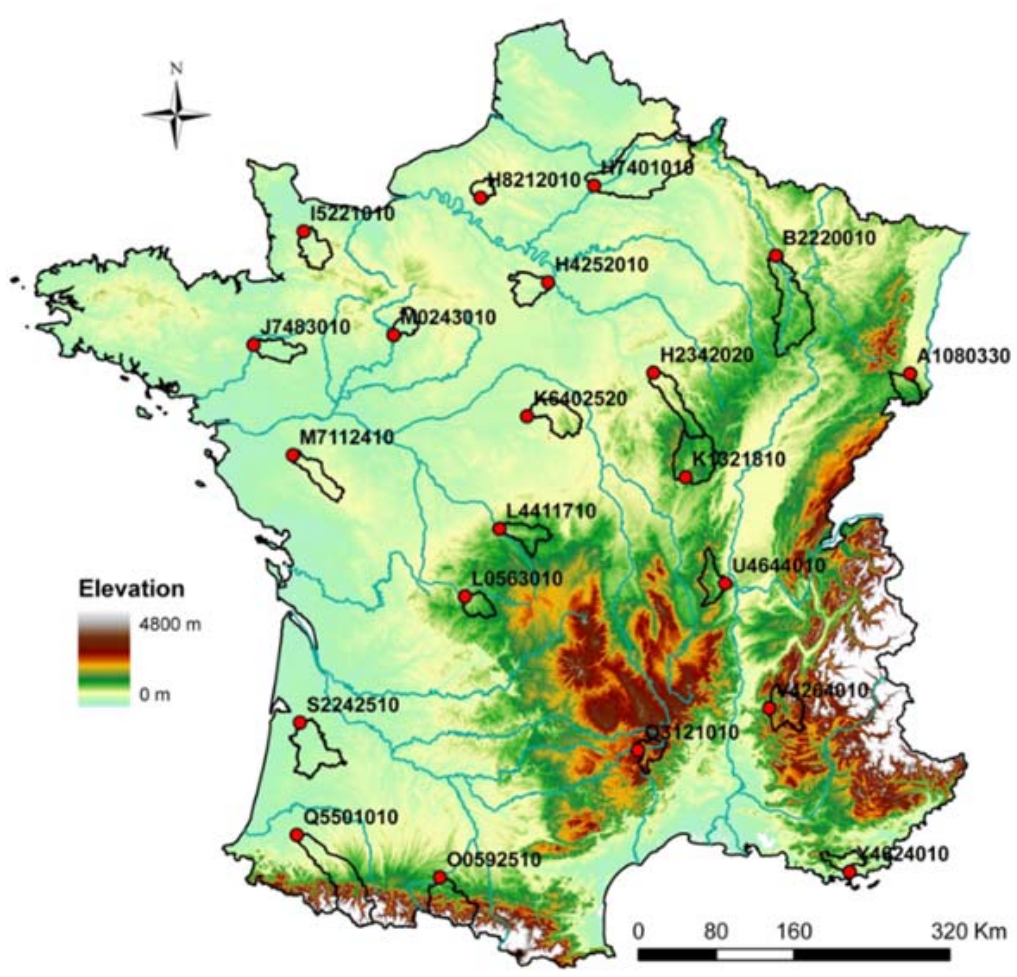

**Figure 3. Location of the 21 catchments in France. Red dots represent the catchment outlets**

## 3.2 Hydrological model

The RAT diagnostic framework is generic and can be applied to any type of model. Here daily streamflow was simulated using the daily lumped GR4J rainfall–runoff model (Perrin et al., 2003). The objective function used for calibration is the KGE criterion (Gupta et al., 2009) computed on square-root-transformed flows. Model implementation was done with the airGR R package (Coron et al., 2017, 2018).

### 3.3 Evaluation of the RAT framework

The RAT was evaluated against the GSST of Coron et al. (2012) used as a benchmark, in order to check whether it yields similar results. The GSST procedure was applied to each catchment using a 10-year period to calibrate the model. For each calibration, each 10-year sliding period over the remaining available period, strictly independent of the calibration one, was used to evaluate the model. The results of the two approaches were compared by plotting on the same graph the annual streamflow bias obtained from the unique simulation period for the RAT, and the average streamflow bias over the sliding calibration-validation time periods for GSST, as a function of temperature, precipitation and humidity anomalies as in Figure 2. The similarity of the trends (between streamflow bias and climatic anomaly) obtained by the two methods was evaluated on the catchment set by comparing the slope and intercept of the linear regressions obtained in each case.

We then identified the catchments where the RAT procedure detected a lack of dependency of streamflow bias to climate variables, or a dependency to one or several variables. The Spearman correlation between model bias and climate variables was computed and a significance threshold of 5% was used (p-value 0.05).

# 4 Results

## 4.1 Comparison between the RAT and the GSST procedure

Figure 4 presents an example for the Orge River at Morsang-sur-Orge: GSST points are represented by black dots and RAT points by red squares. Let us first note that since red points represent only each of the N years of the period for the RAT and black points represent all GSST possible independent calibration-validation pairs (a number close to *N(N-1)*), black points are much more numerous. We can observe that the amplitude of both streamflow bias and climatic variable change is larger for the GSST than for the RAT as there are more calibration periods, whatever the climatic variable (P, T or HI). However, the trends in the scatterplot are quite similar. Graphs for all catchments are provided as supplementary material.

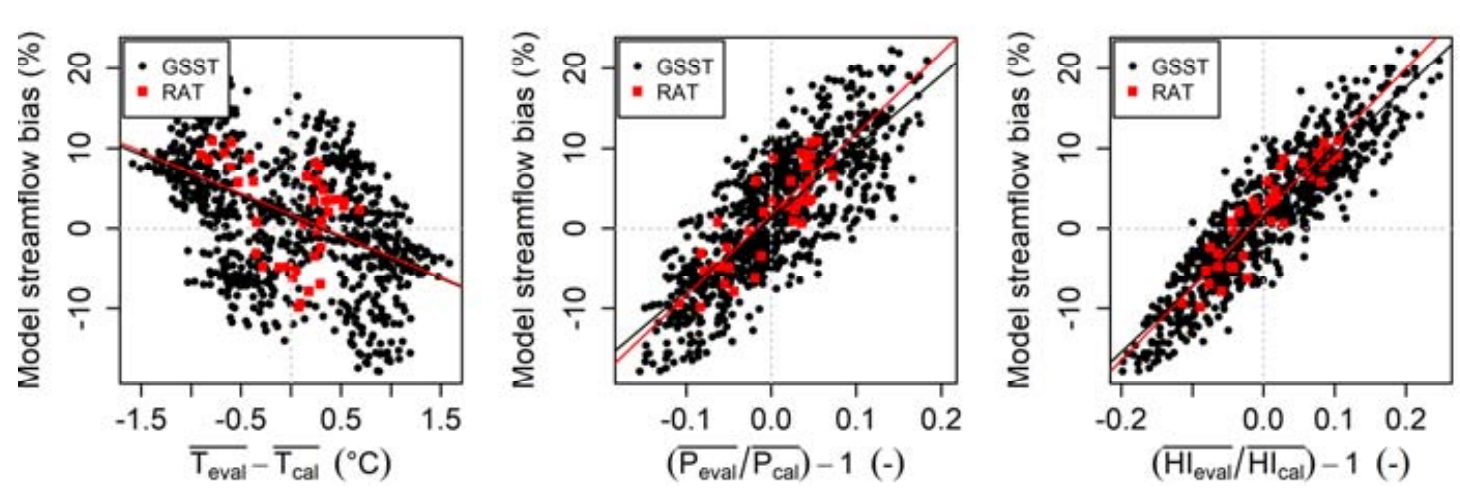

**Figure 4. Streamflow bias obtained with the RAT (red squares) and the GSST (black dots), as a function of temperature, precipitation and humidity index anomalies, for the Orge River at Morsang-sur-Orge (H4252010) (934 km²).**

To summarize the results on the 21 catchments, we present on Figure 5 the slope and intercept of a linear regression computed between model streamflow bias and climatic variable anomaly, for the

GSST and the RAT over the 21 catchments: the slope of the regressions obtained for both methods are very similar and the intercept also exhibits a good match (although somewhat larger differences).

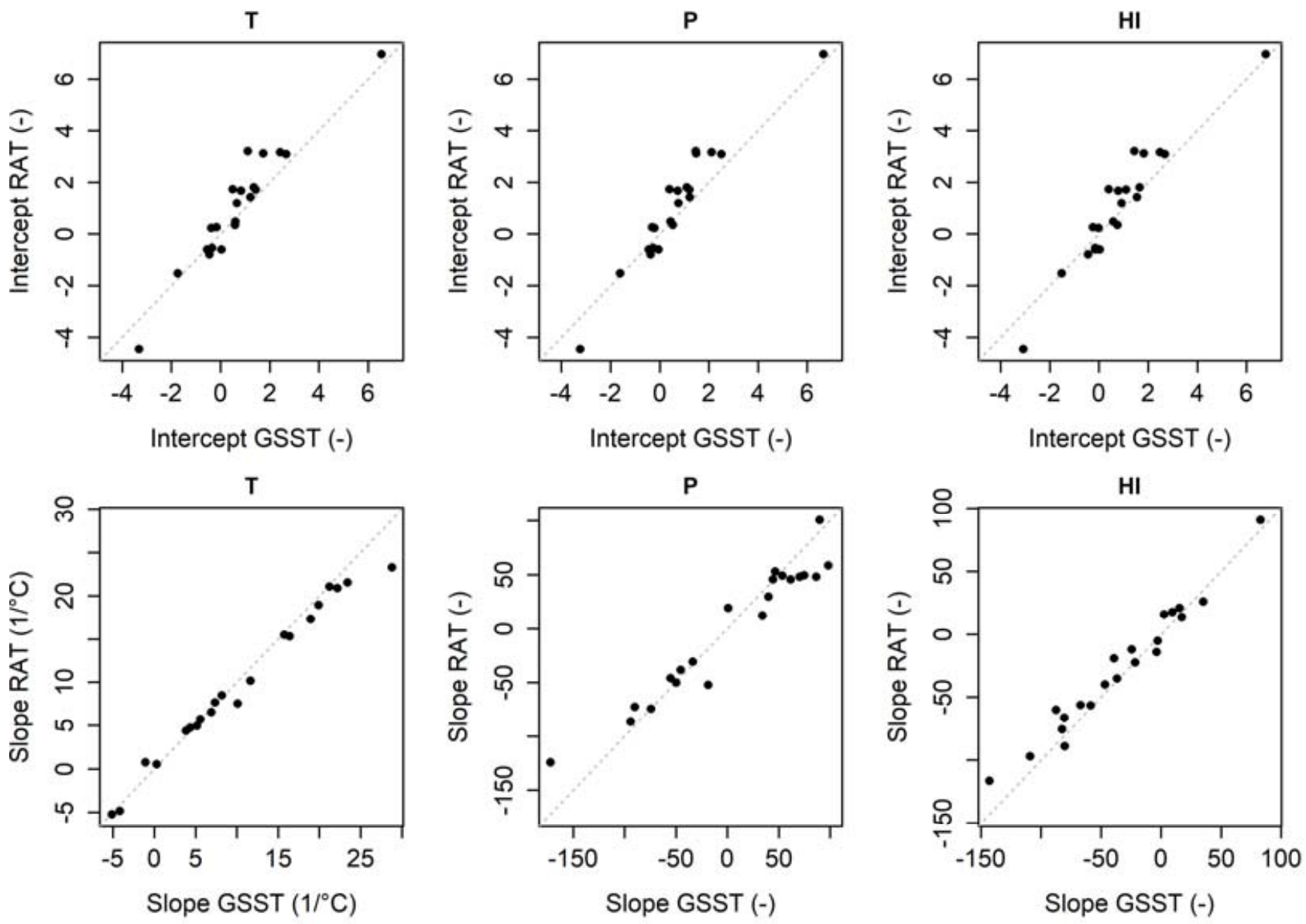

**Figure 5. Comparison of slopes and intercept of linear regressions between streamflow bias and temperature (T), precipitation (P) and humidity index (HI) anomalies (from left to right) obtained by the GSST and the RAT procedures (each point represents one of the 21 test catchments)**

We can thus conclude that the RAT reproduces the results of GSST, but at a much lower computational price, and this is what we were aiming at. One should however acknowledge that switching from the GSST to the RAT unavoidably reduces the severity of the climate anomalies we can expose the hydrological models to: indeed, the climate anomalies with the RAT are computed with respect to the mean over the whole period, whilst with the GSST they are computed between two shorter (and hence potentially more different) periods.

## 4.2 Application of the RAT procedure to the detection of climate dependencies

We now illustrate the different behaviours found among the 21 catchments when applying the RAT procedure. The significance of the link between model bias and climate anomalies was based on the Spearman correlation and a 5% threshold. Five cases were identified:

1. **No climate dependency** (Figure 6): This is the case for 6 catchments out of 21 and the expected situation of a "robust" model. The different plots show a lack of dependence, for temperature, precipitation and humidity index alike. For the catchment of Figure 6, the p-value of the Spearman correlation is quite high (between 0.23 and 0.98) and thus not significant.

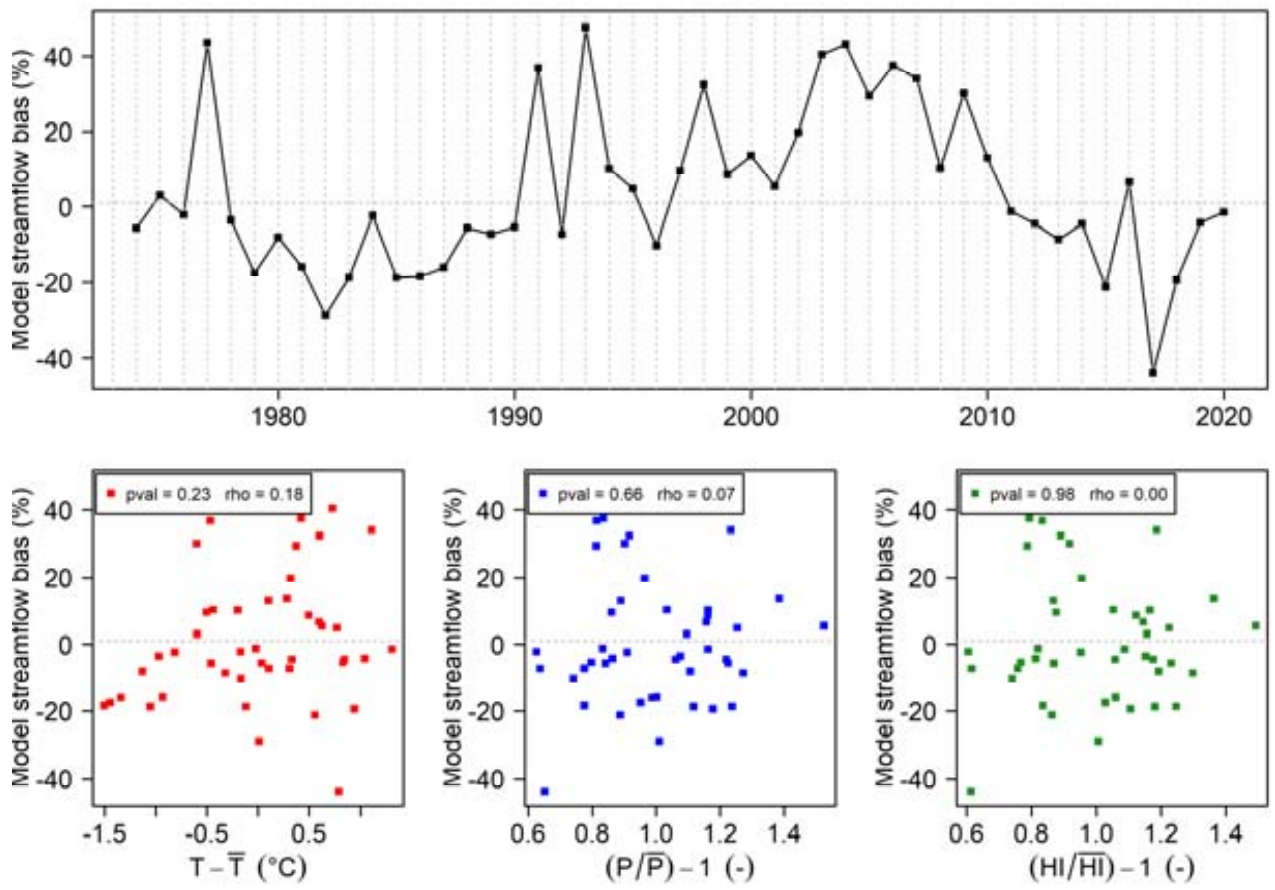

**Figure 6. Streamflow annual bias obtained with the RAT function of time (top), temperature absolute anomalies (bottom left), and precipitation P (bottom centre) and humidity index $P/E_0$ (bottom right) anomalies, for the Orne Saosnoise River at Montbizot (M0243010) (510 km²).**

2. **Significant dependency on annual temperature, precipitation and humidity index** (Figure 7): This is a clearly undesirable situation illustrating a lack of robustness of the hydrological model. It happens on only two catchments out of 21. The Spearman correlation between model bias and temperature, precipitation and humidity index anomalies (respectively 0.49, -0.36 and -0.46) is significant (i.e. below the classic significance threshold of 5%). In Figure 7, the annual bias shows an increasing trend with annual temperature and a decreasing trend with annual precipitation and humidity index.

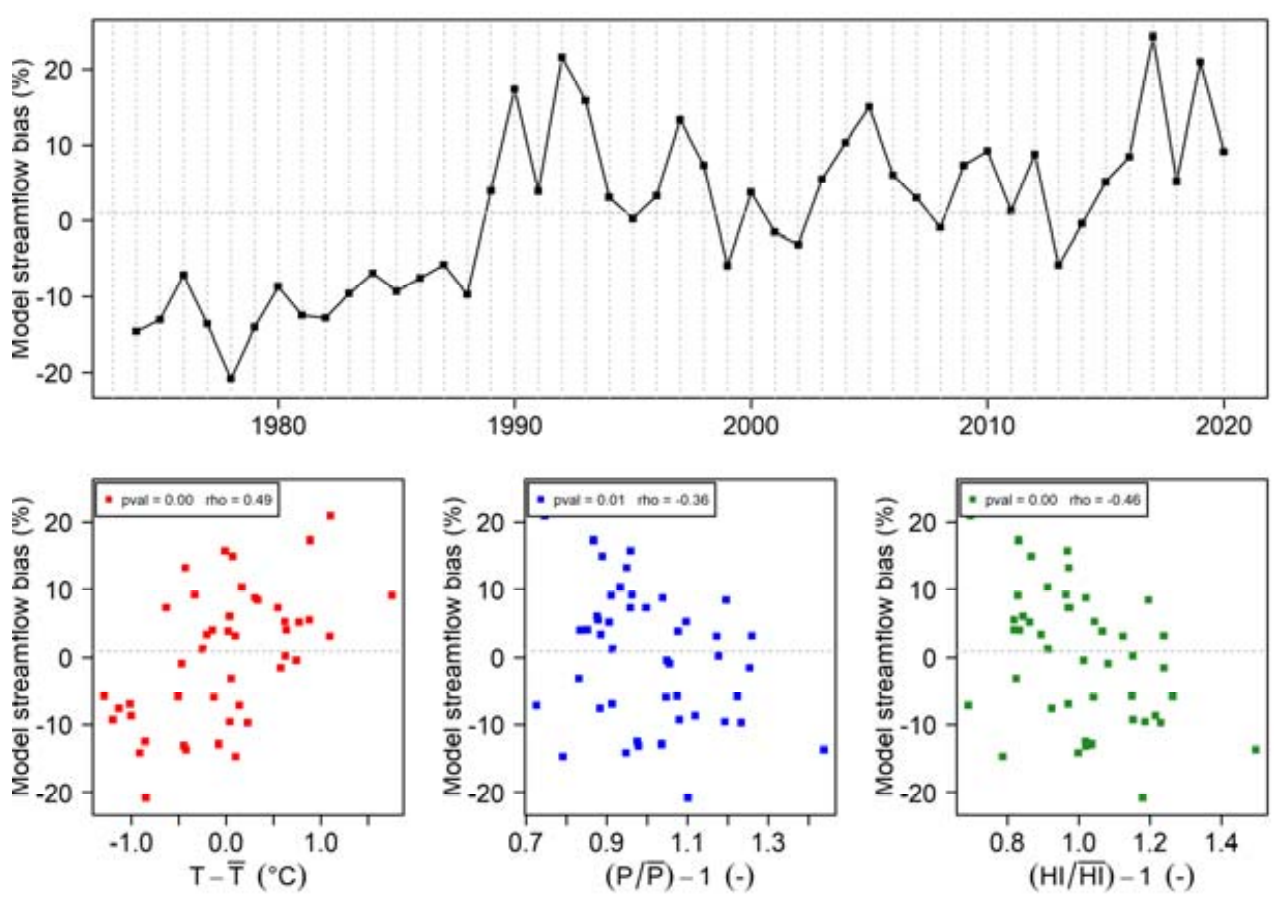

**Figure 7. Streamflow annual bias obtained with the RAT function of time (top), temperature absolute anomalies (bottom left), and precipitation P (bottom center) and humidity index $P/E_0$ (bottom right) anomalies, for the Arroux River at Etang-sur-Arroux (K1321810) (1790 km²)**

3. **Significant climate dependency on precipitation and humidity index but not on temperature** (Figure 8). This case happens on 5 of the 21 catchments.

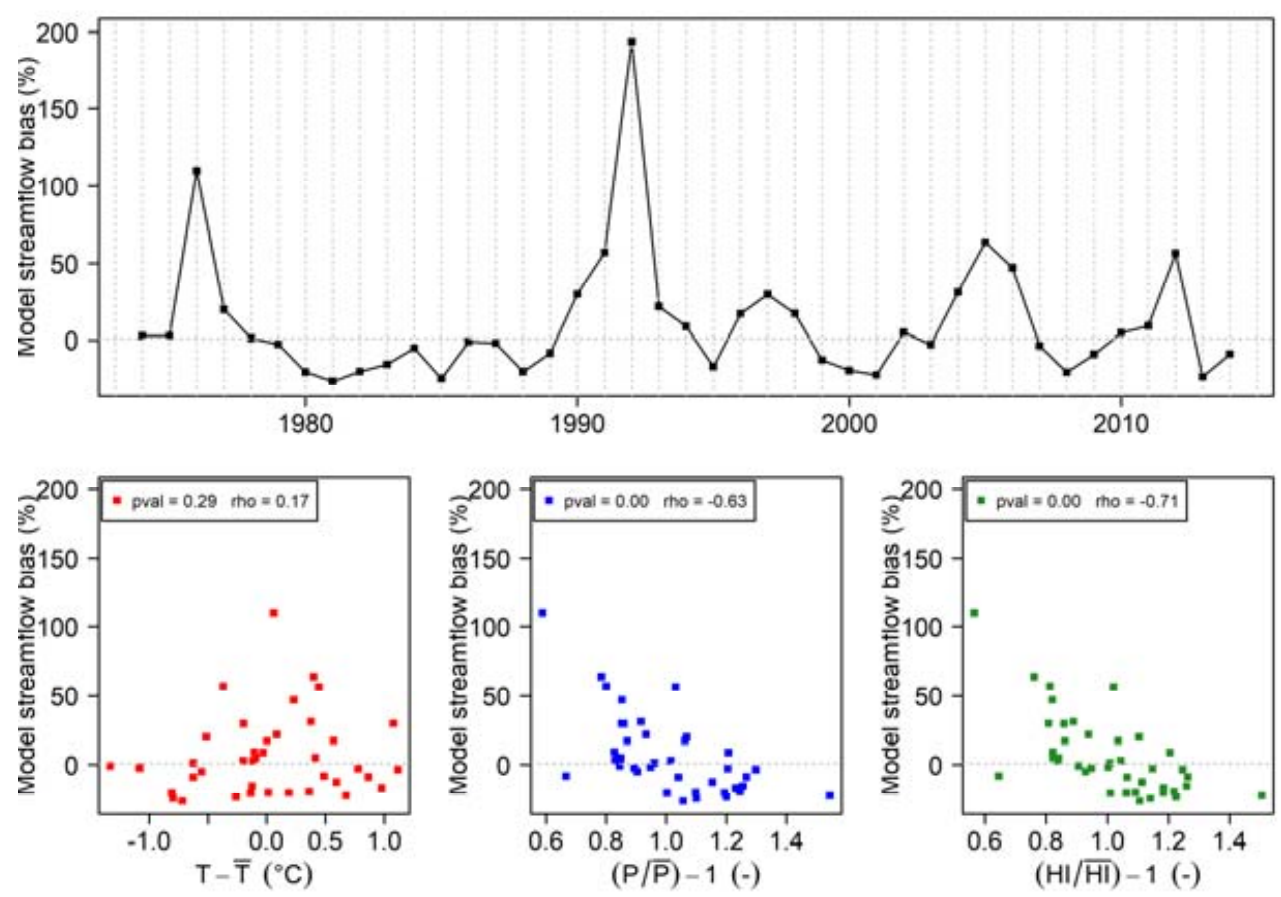

**Figure 8. Streamflow annual bias obtained with the RAT function of time (top), temperature absolute anomalies (bottom left), and precipitation P (bottom center) and humidity index P/$E_0$ (bottom right) anomalies, for the Seiche River at Bruz (J7483010) (810 km²)**

4. **Significant climate dependency on temperature but not on precipitation and humidity index** (Figure 9). This case happens on 3 of the 21 catchments.

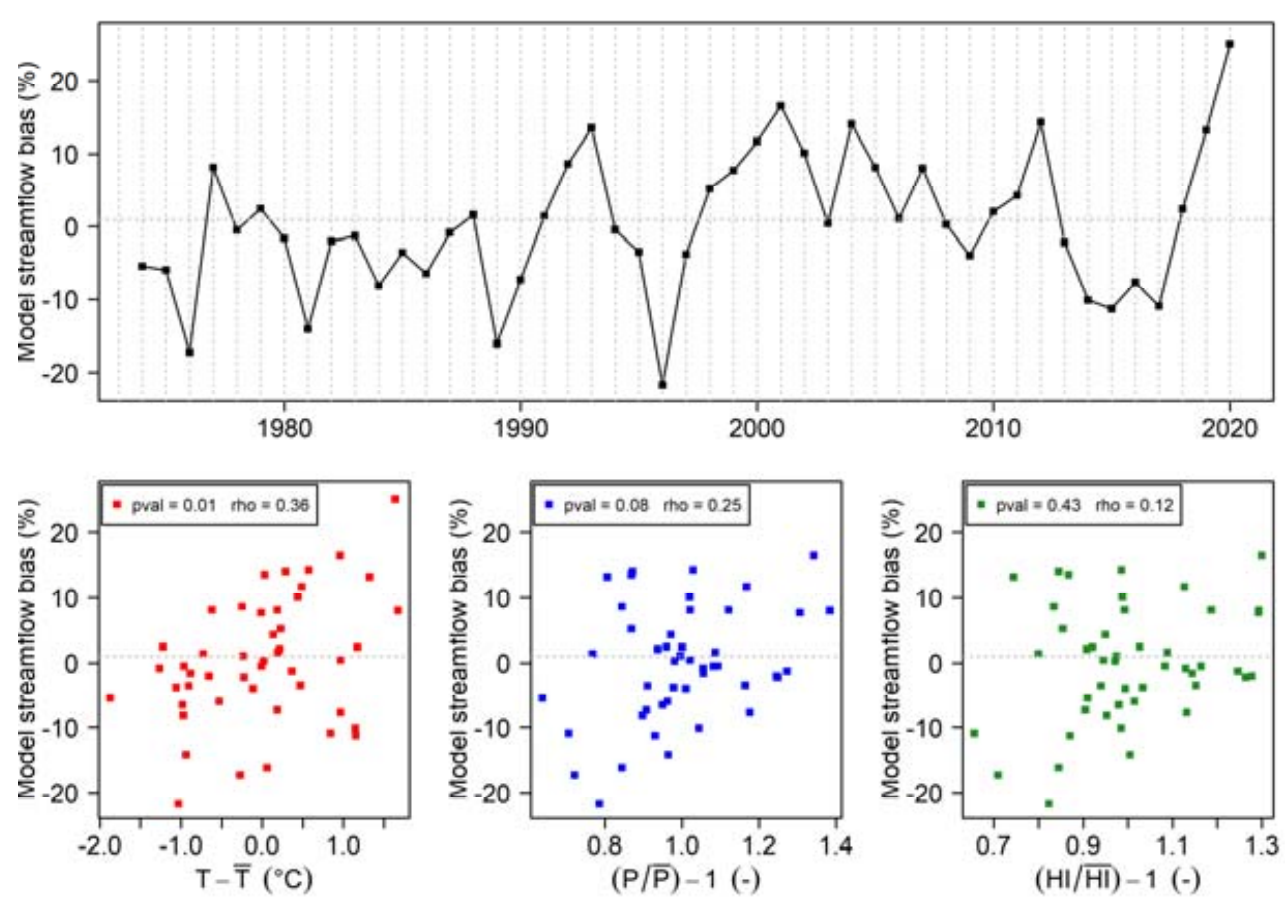

**Figure 9. Streamflow annual bias obtained with the RAT function of time (top), temperature absolute changes (bottom left), and precipitation P (bottom center) and humidity index P/$E_0$ (bottom right) anomalies, for the Ill at Didenheim (A1080330) (670 km²)**

5. **Significant climate dependency on temperature and humidity index but not on precipitation** (Figure 10). This case happens on 5 of the 21 catchments.

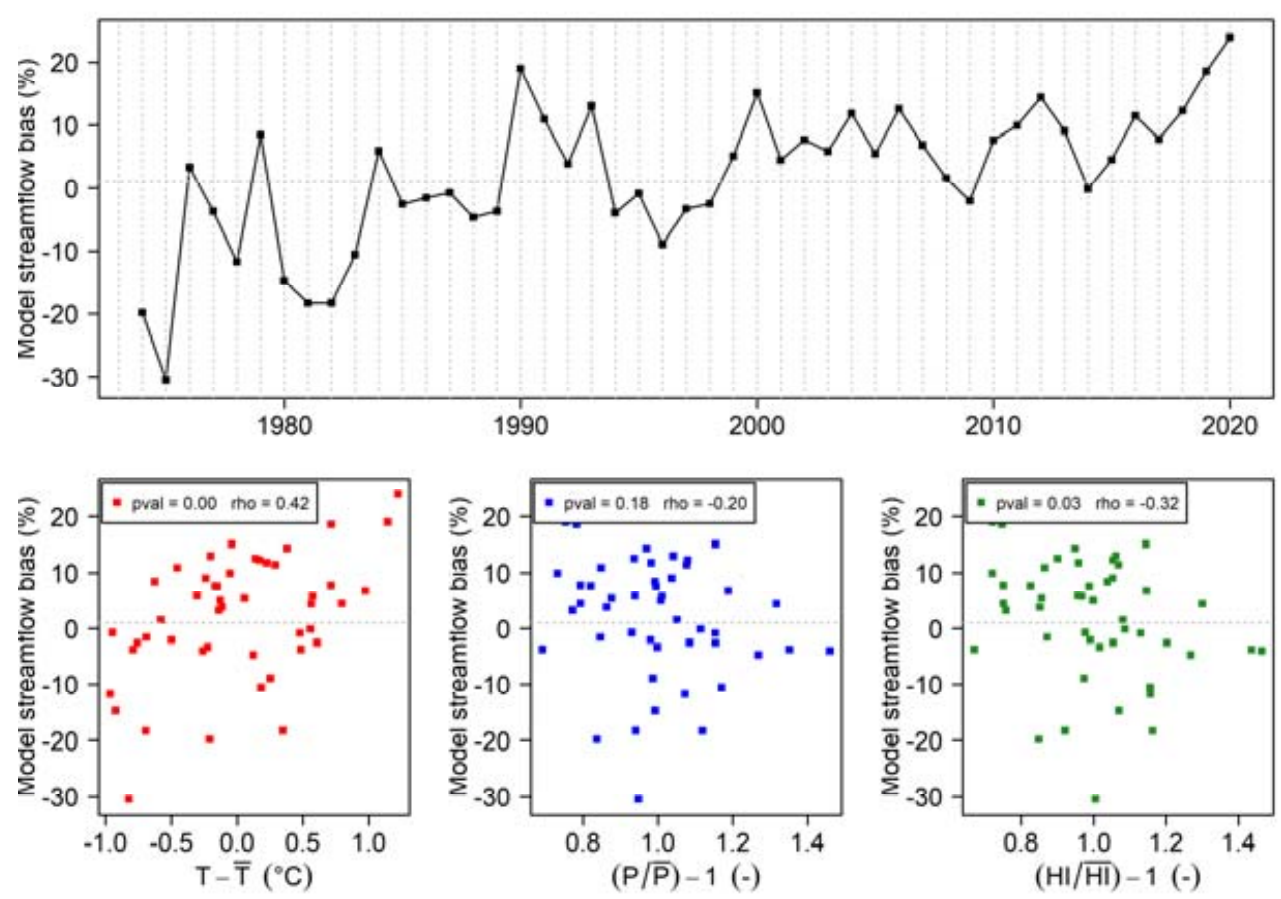

**Figure 10. Streamflow annual bias obtained with the RAT function of time (top), temperature absolute changes (bottom left), and precipitation P (bottom center) and humidity index P/$E_0$ (bottom right) anomalies, for the Briance River at Condat-sur-Vienne (L0563010) (597 km²)**

## 5 Conclusion

The proposed robustness assessment test (RAT) is an easy-to-implement evaluation framework that allows robustness evaluation from all types of hydrological models to be compared, by using only one long period for which model simulations are available. The RAT consists in identifying undesired dependencies of model errors to the variations of some climate variables over time. Such dependencies can indeed be detrimental for model performance in a changing climate context. This test can be particularly useful for climate change impact studies where the robustness of hydrological models is often not evaluated at all: as such, our test can help users to discriminate alternative models and select the most reliable models for climate change studies, which ultimately should reduce uncertainties on climate change impact predictions (Krysanova et al., 2018).

The proposed test has obviously its limits, and a first difficulty that we see in using the RAT is that it is only applicable in cases where the hypothesis of independence between the 1-year subperiods and the whole period is sufficient. This is the case when long series are available (at least 20 years, see last graph in appendix). If it is not the case, the RAT procedure should not be used. Therefore, we would indeed recommend its use in cases where modellers cannot “afford” multiple calibrations, or where the parameterisation strategy is considered (by the modeller) as ‘calibration free’ (i.e. physically-based models). A few other limitations should be mentioned:

1. In this note, the RAT concept was illustrated with a rank-based test (Spearman correlation) and a significance threshold of 0.05. Like all thresholds, this one is arbitrary. Moreover, other non-parametric tests could be used and would probably yield slightly different results (we also tested the Kendall tau test, with very similar results, but do not show the results here);
2. Detecting a relationship between model bias and a climate variable using the RAT does not allow to directly conclude on a lack of model robustness. Indeed, changes in the precipitation monitoring network or in the hydrometric rating curves can also give the false impression that the hydrological model lacks robustness. Such an erroneous conclusion could also be due to

widespread changes in land use, construction of an unaccounted storage reservoir or the evolution of water uses. Some of the lacks of robustness detected among the 21 catchments presented here could be in fact due to metrological causes;

3. Also, because of the ongoing rise of temperatures (over the last 40 years at least), we have a correlation between temperature and time since the beginning of streamgaging. If for any reason, time is having an impact on model bias, this may cause an artefact in the RAT in the form of a dependency between model bias and temperature;
4. Similarly to the Differential Split Sample Test, the diagnostic of model climatic robustness is limited to the climatic variable against which the bias is compared. As such, the RAT should not be seen as an *absolute* test, but rather as a *necessary but not sufficient* condition to use a model for climate change studies: because the climatic variability present in the past observations is limited to the historic range, so is the extrapolation test. With Popper's words (Popper, 1959), the RAT can only allow falsifying a hydrological model… but not proving it true;
5. Although it would be tempting to transform the RAT into a post-processing method, we do not recommend it. Indeed, detecting a relationship between model bias and a climate variable using the RAT does not necessarily mean that a simple (linear) debiasing solution can be proposed to solve the issue (see e.g. the paper by Bellprat et al. (2013) on this topic). What we do recommend is to work as much as possible on the model structure, to turn it less climate dependent;
6. Last, we could mention that a model showing a small overall annual bias (but linked to a climate variable) could still be preferred to one showing a large overall annual bias (but independent of the tested climate variables): the RAT should not be seen as the only basis for model choice.

Beyond the limitations, we also see the perspective for further development of the method: although this note only considered overall model bias (as the most basic requirement for a model to be used to predict the impact of a future climate), we think that this methodology could be applied to bias in different flow ranges (low or high flows) or to statistical indicators describing low-flow characteristics or maximum annual streamflow. And characteristics other than bias could be tested, e.g. ratios pertaining to the variability of flows. Further, while we only tested the dependency to mean annual temperature, precipitation and humidity index, other characteristics, such as precipitation intensity or fraction of snowfall, could be considered in this framework.

# 6 Acknowledgments

This work was funded by the project AQUACLEW, which is part of ERA4CS, an ERA-NET initiated by JPI Climate, and funded by FORMAS (SE), DLR (DE), BMWFW (AT), IFD (DK), MINECO (ES), ANR (FR) with co-funding by the European Commission [Grant 690462].

The authors gratefully acknowledge the comments of Prof. Jens-Christian Refsgaard and Dr Nans Addor on a preliminary version of the note, as well as the help of Dr Léonard Santos for the flowchart.

The gridded SAFRAN climate reanalysis data can be ordered from Météo-France.

Observed streamflow data are available on the French HYDRO database (http://www.hydro.eaufrance.fr/).

The GR models, including GR4J, are available from the airGR *R* package.

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

# 8 Appendix – Checking the impact of the partial overlap between calibration and validation periods in the RAT

In this appendix, we deal with calibrated models, for which we verify that the main hypothesis underlying the RAT is reasonable, i.e. that when considering a long calibration period, the weight of each individual year in the overall calibration process is almost negligible. We then explore the limits of this hypothesis when reducing the length of the overall calibration period.

- **Evaluation method**

In order to check the impact of the partial overlap between calibration and validation periods in the RAT, it is possible (provided one works with a calibrated model) to compare the RAT with a "leave-one out" version of it, which is a classical variant of the Split Sample Test (SST): instead of computing the annual bias after a single calibration encompassing the whole period (RAT), we compute the annual bias with a different calibration each time, encompassing the whole period minus the year in question ("leave-one-out SST").

The comparison between the RAT and the SST can be quantified using the root mean square difference (RMSD) of annual biases:

$$RMSD_{Bias} = \sqrt{\left(Bias_{RAT} - Bias_{SST}\right)^2} \qquad \text{Eq.1}$$

where $Bias_{RAT}$ is the bias of validation year *n* when calibrating the model over the entire period (RAT procedure), and $Bias_{SST}$ the bias of validation year *n* when calibrating the model over the entire period minus year *n* (leave-one-out SST procedure).

The difference between the two approaches is schematized in Figure 11: the leave-one-out procedure consists in performing *N* calibrations over (*N-1*)-year-long periods followed by an independent evaluation on the remaining 1-year-long period. As shown in Figure 11, the two procedures result in the same number of validation points (*N*). Eq. 1 provides a way to quantify whether both methods differ, i.e. whether the partial overlap between calibration and validation periods in the RAT makes a difference.

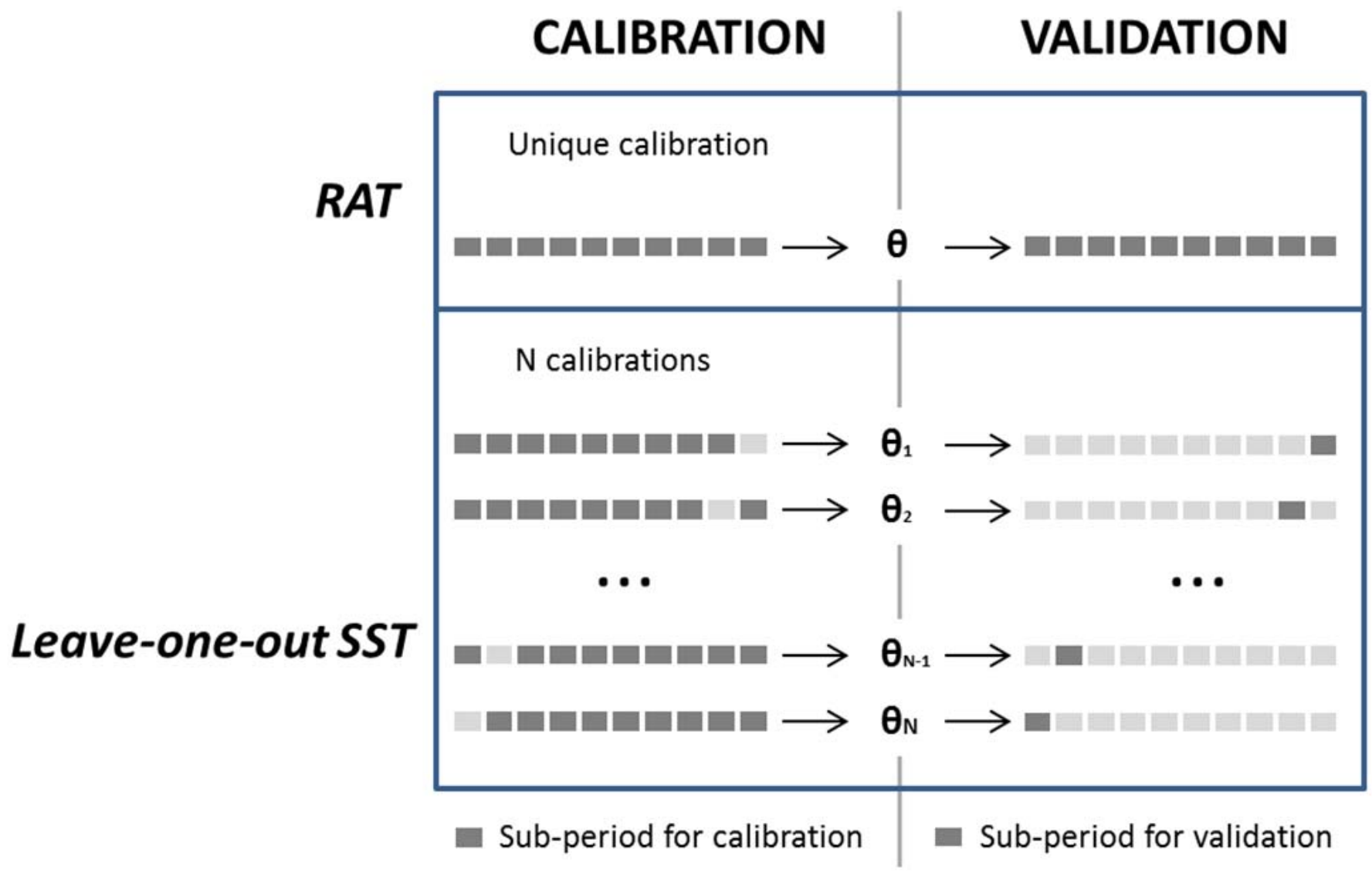

**Figure 11. Comparison of the RAT procedure with a leave-one-out split-sample test (SST). Both methods have N validation periods (one per year). The RAT needs only one calibration, whereas the SST requires N calibrations. Dark grey squares represent the years used for calibration or validation**

- **Comparison between the RAT and the leave-one-out SST**

Figure 12 plots the annual bias values obtained with the RAT versus the annual bias obtained with the leave-one-out SST for the 21 test catchments, showing a total of 21x47 points. The almost perfect alignment confirms that our underlying "negligibility" hypothesis is reasonable (at least on our catchment set).

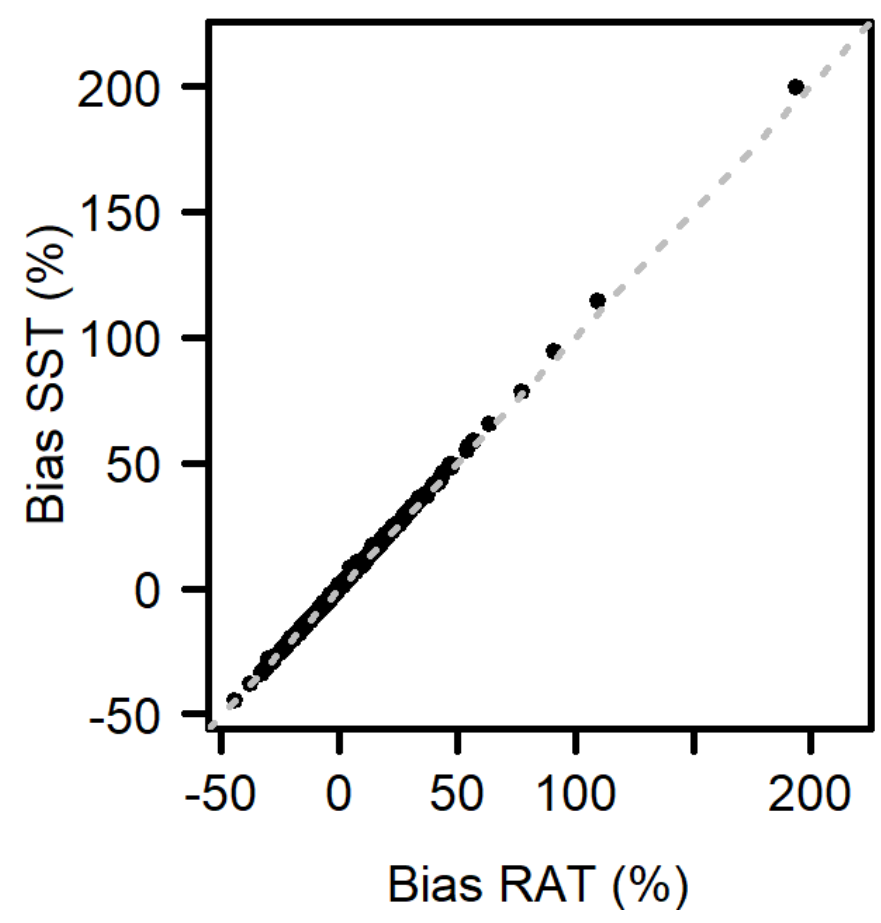

**Figure 12. Comparison of the annual bias obtained with the RAT with the annual bias obtained with the leave-one-out SST. Each of the 21 catchments is represented with annual bias values (47 points by catchment, 21x47 points in total)**

Figure 13 presents the Spearman correlation p-values for the correlation between annual bias and changes in annual temperature, precipitation, and humidity index ($P/E_0$), for the RAT and the leave-one-out SST. The results from the RAT and the SST show the same dependencies on climate variables (similar *p*-values).

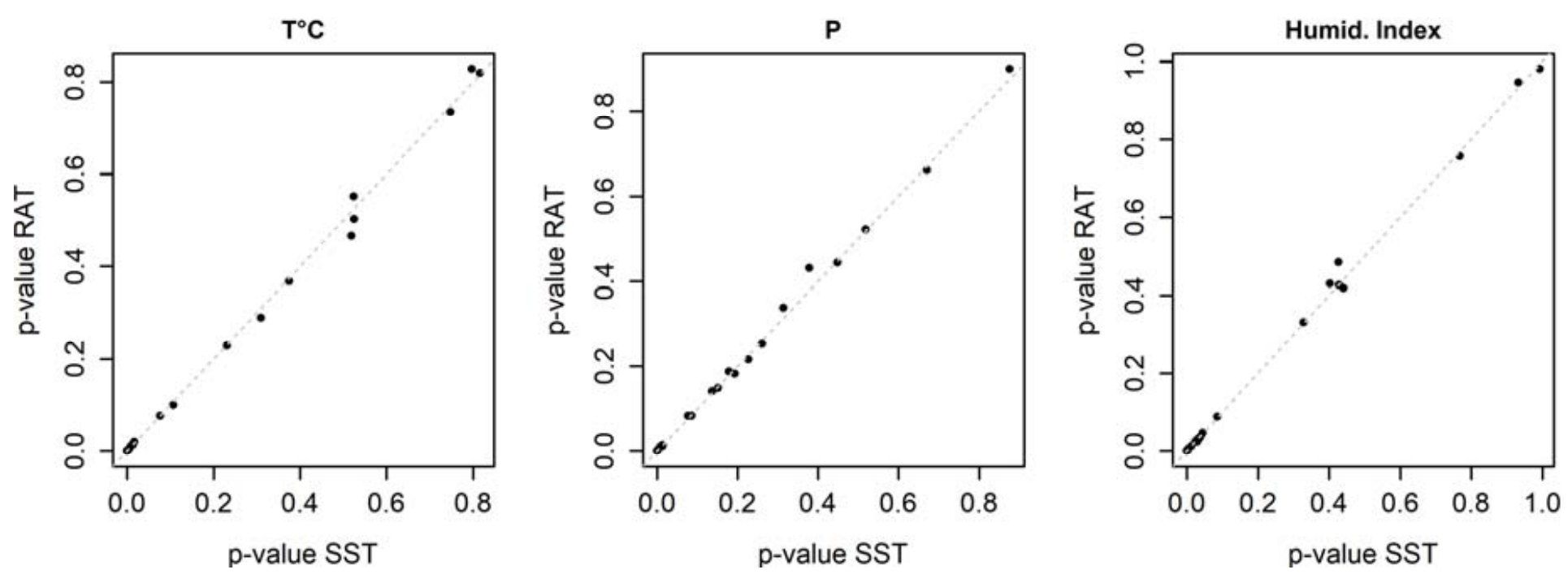

**Figure 13. Spearman correlation *p*-value from the correlation for annual bias and annual temperature, precipitation, and humidity index ($P/E_0$). Comparison between RAT and SST (one point per catchment)**

- **Sensitivity of the RAT procedure to the period length**

It is also interesting to investigate the limit of our hypothesis (i.e. that the relative weight of one year within a long time series is very small) by progressively reducing the period length: indeed, the shorter the data series available to calibrate the model, the more important the relative weight of each individual year. Figure 14 compares the annual bias obtained with the RAT procedure with the annual bias obtained with the leave-one-out SST, for 10-, 20-, 30-, and 40-year period lengths (selection of the shorter periods was realized by sampling 10, 20, 30, and 40 years regularly among the complete time series). The shorter the calibration period, the larger the differences between both approaches (wider points scatter): there, we reach the limit of the single calibration procedure. We would not advise to use RAT with time series of less than 20 years.

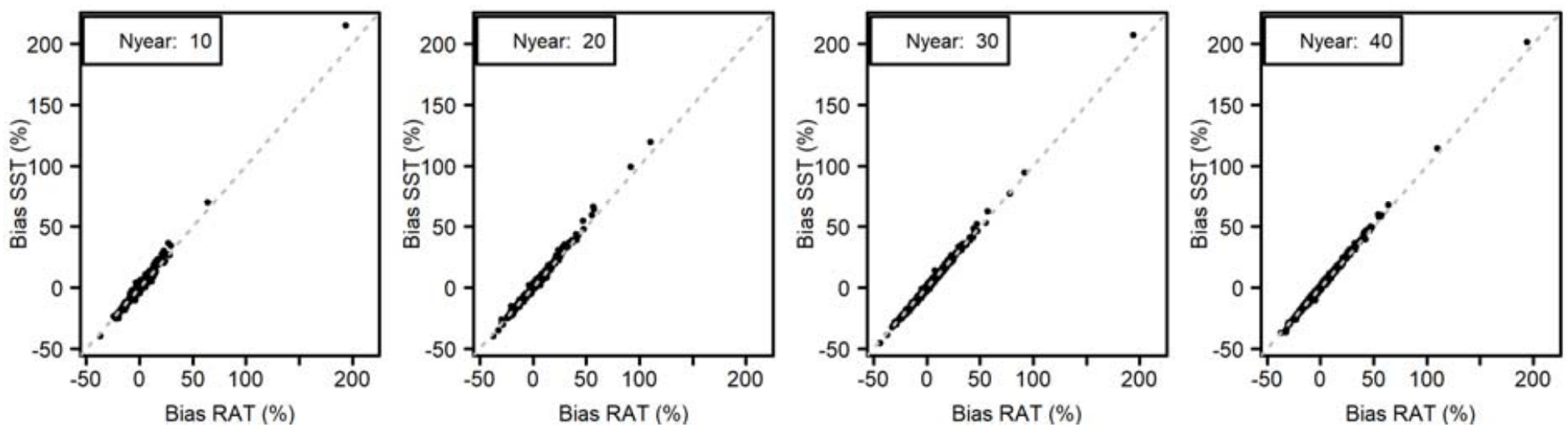

**Figure 14. Annual bias obtained with the RAT procedure vs. annual bias obtained with leave-one-out SST. Shorter time periods are obtained by sampling 10, 20, 30, and 40 years regularly among the complete time series. Each of the 21 catchments is represented with annual bias values**

These differences can be quantitatively measured by computing the RMSD (see Eq.1) between the annual bias obtained with the RAT procedure and with the SST for different calibration period lengths (see Figure 15). The RMSD tends to increase when the number of years available to calibrate the model decreases, but it seems to be stable for periods longer than 20 years.

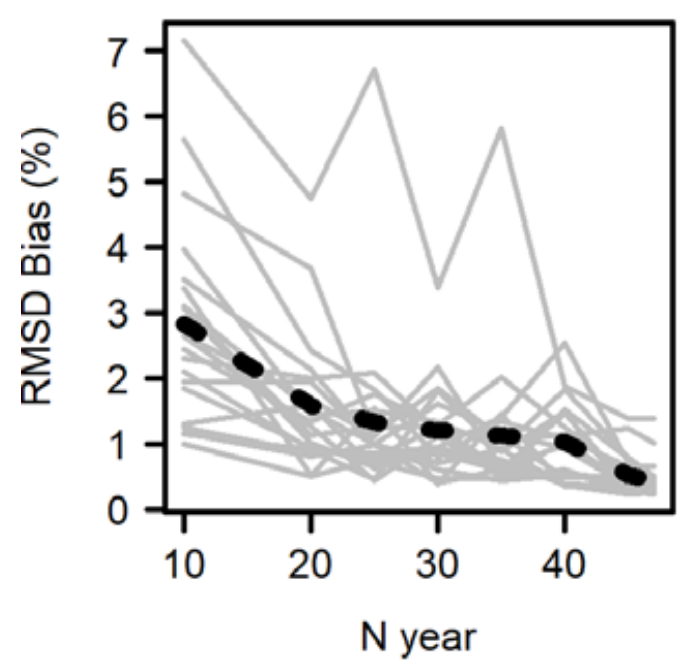

**Figure 15. RMSD between annual bias obtained with the RAT procedure and with the leave-one-out SST for different calibration period lengths for each catchment. The dotted line represents the mean RMSD for all catchments. Each grey line represents one of the 21 catchments.**