# Peer review of "Technical Note – RAT: a Robustness Assessment Test for calibrated and"

_Hydrology and Earth System Sciences, 2021_

## Author Comment (AC3)

Thank you very much for your review. In addition to our general answer, we provide here specific answers to the points you raised (in blue while your comments remain in black).

**1. General comments**

This technical note by Nicolle et al. presents an evaluation method for hydrological model robustness and is called the Robustness Assessment Test (RAT). The main assumption of the RAT is that the bias in the model output (i.e. streamflow) should not be correlated with the climatic input data (e.g. precipitation, temperature, air humidity), in which case, the model has a dependency on climactic variables, thereby not suitable for use in climate change impact studies. The manuscript is relatively well structured and written. The development of a framework for hydrological model evaluation is relevant, and can potentially be of interest for the readers of HESS if the RAT is thoroughly evaluated and some of its limitations are addressed. The RAT seems to be developed to detect deficiencies in model structure but it is not clearly demonstrated that model rejection by the RAT is not also due to input data. This is a key point that the authors have to address. Please refer to my comments and suggestions below.

**2. Specific comments**

**2.1 Major Comments**

**2.1.1 Contradictions**

The main assumption is that there should not be a dependency between model output bias and the input climatic variable for a hydrological model to be considered robust (Lines 152-157). However, in the conclusion, the authors clearly recognize that "Detecting a relationship between model bias and a climate variable using the RAT does not allow to directly conclude on a lack of model robustness". They further mention in the key points that "success at the RAT test is a necessary (but not sufficient) condition of model robustness". Therefore, I wonder why the RAT should be used for model evaluation.

We consider that given that the input data are reliable (which is to be checked before any study), a fail in passing the RAT is very likely indicating a model robustness issue. We will rephrase the mentioned sentences to avoid any misunderstanding.

**2.1.2 Input data**

How do you know if the model failure at the RAT is due to the dependency on input data or to the dependency on model structure or parametrization?

We assume that the reviewer means here "dependency on input data bias", not solely "dependency on input data". As mentioned below, we consider that checking the quality of input data has to be done prior to assessing the robustness of models. The RAT does not allow to decipher the sources of these dependencies.

L170-171: Is the model rejected because the parametrization is wrong or because the input data is wrong? Would you get the same results with a different precipitation data? A clear answer to this question is essential for this work, as we do not want to reject a model that is not wrong.

We agree with you that "we do not want to reject a model that is not wrong", but like with any test, this risk (type I error) will exist. This is the risk of judging guilty an innocent person, which, in the case of a model is not so worrying: it is in the very nature of hydrological models to be guilty or imperfect. A test such as

RAT may yield a false positive for a few catchments, but it will nonetheless offer a possibility for comparing modelling alternatives.

In their work, the authors assume that in case there is a correlation between the model output bias and the input data, that correlation is due to the model structure/parametrization, as the authors do not investigate the potential contribution of the input data to the model output bias. This is a clear limitation of the evaluation of the RAT method. Without testing different input data, the authors implicitly assume that the only input data that they are using is right or at least is not the source of errors in the model outputs. But we know that input data remain a main source of uncertainty in hydrological modelling (Gupta, 1998 https://agupubs.onlinelibrary.wiley.com/doi/abs/10.1029/97WR03495). The authors recognize in the conclusion that the lack of model robustness can also be due to meteorological causes (L324-325). I urge the authors to test different input datasets, which are largely available nowadays because of the availability of satellite and reanalysis products. Different precipitation and temperature datasets must be tested to demonstrate that model rejection based on the RAT is not false negative, i.e. Type 2 error that we want to avoid in model evaluation (Beven, 2010 https://onlinelibrary.wiley.com/doi/epdf/10.1002/hyp.7718).

We do agree on the rationale of these comments. However, we believe that addressing data quality in this paper would deviate the aim of this work, which is presenting a method to assess model robustness. Also, we think that global datasets will be unavoidably much more imprecise than Meteo-France's ground-based interpolated data, which will increase considerably the amplitude of bias (we have a long experience of using Meteo France's SAFRAN product as catchment forcing, and we consider it to be the best option in France).

Another option to test if the model performance depends on precipitation, for instance, is to take a precipitation product and gradually apply some perturbation of [-30, -20, -10, 0, 10, 20, 30] percent bias and check if the bias in streamflow is correlated to precipitation for the different scenarios.

This suggestions is interesting, but the data quality issues that modellers are the most concerned with are not random errors, they are the trends or the sudden changes linked with modifications of the observation network (we could still add noises with a trend but it would be a rather obvious result that these are detectable).

**2.1.3 Seasonality**

Is the RAT valid for catchments with a strong seasonality in climatic variables? In case of bias in the input data, wouldn't that easily be reflected in the model output for catchments with strong seasonality? Thereby, misleading the conclusion of the RAT as the RAT would reject the model assuming that it is its parametrization that is problematic?

We believe that the RAT is still valid for catchments with a strong seasonality. However, we recognize seasonal biases if they do compensate each other would not be detected by the RAT in its present form. A seasonal bias could indeed help.

**2.1.4 Annual aggregation - Hydrological year**

L159: Fig.1 - What is the reasoning behind the annual aggregation of the data by hydrological year to compute the bias? What are the implications of the choice of the hydrological year on the calculation of the bias?

You are right, a perfect model does not need a specific season to compute bias. But in the case of snow-affected catchments, it seems always more careful to avoid separating the snow accumulation and the snow melt seasons. We applied the RAT both with calendar and hydrological years, and obtained a difference on only one catchment (see table below).

*Table 1. Comparison of RAT results depending of the time step chosen for bias calculation (calendar year vs. hydrological year)*

| Catchment | Significant dependency with calendar years | Significant dependency with hydrological years |
|---|---|---|
| A1080330 | FALSE | TRUE |
| B2220010 | TRUE | TRUE |
| H2342020 | TRUE | TRUE |
| H4252010 | TRUE | TRUE |
| H7401010 | FALSE | FALSE |
| H8212010 | TRUE | TRUE |
| I5221010 | TRUE | TRUE |
| J7483010 | TRUE | TRUE |
| K1321810 | TRUE | TRUE |
| K6402520 | TRUE | TRUE |
| L0563010 | TRUE | TRUE |
| L4411710 | TRUE | TRUE |
| M0243010 | FALSE | FALSE |
| M7112410 | FALSE | FALSE |
| O0592510 | FALSE | FALSE |
| O7101510 | TRUE | TRUE |
| Q5501010 | TRUE | TRUE |
| S2242510 | FALSE | FALSE |
| U4644010 | TRUE | TRUE |
| V4264010 | TRUE | TRUE |
| Y4624010 | FALSE | FALSE |
| **TOTAL** | **14** | **15** |

The model output bias and climatic variables may not be dependent at daily scale but show dependency at coarser temporal scale, or vice versa. How would that be captured by the RAT? The RAT might reject the model at annual scale while there is no dependency at daily scale (temporal resolution of the hydrological simulations), or inversely.

We consider that because hydrological models are full of imperfections, it is more reasonable to work on bias at coarse time steps first, and then only to move to fine time steps. But we agree that the absence of bias for any given year, may well hide a positive bias during half of the year and a negative year during the other half. This is why we underline in conclusion that "the RAT should not be seen as the only basis for model choice."

**2.2 Minor comments**

L15: I see the RAT as a "complement" rather than an alternative to the "split-sample test".

We disagree on this comment. Indeed, the split-sample test cannot be applied to all models, e.g. physically-based models that do not use calibration. For these models, the RAT is therefore an alternative, not a complement.

A model might pass an SST but fail the RAT, or vice versa. What would happen in those cases? Should the model be rejected or accepted? Which of the SST and RAT outcomes would you give a preference?

This is a tough question, and it also depends on the threshold chosen (the p-value in the case of the RAT, the drop in criterion value between the calibration and the validation period in the case of the SST). Usually, people trust more a method that they have been using for a long time (experience with a method is an important factor of adoption). We have a little more than one year of experience with RAT, much more with SST... However we like the visualization opportunity that the RAT offers, this is why we are tempted to answer that we would give our preference to the RAT results.

L16: "the RAT method does not require multiple calibrations". This sentence can be misinterpreted because it seems like the RAT method could be calibrated while you are referring to the hydrological model. Should be "...multiple calibrations of hydrological models". Also correct this at line 137.

We agree and will make the modifications.

L20: As you said that success at the RAT is a necessary BUT not a sufficient condition of model robustness, can you call your approach a "robustness" assessment test? if a model is robust it would be successful at the RAT, but success at the RAT does not necessarily mean the model is robust. This key point highlights a strong limitation of RAT because you cannot confirm the robustness of the model but just have a hint that it might be robust.

We don't know of perfect tests (there is always a type I and type II error possibility, as you mentioned it). RAT is definitely not a "Robustness Assessment Certificate", but we do think that it is fair to call it a test.

L319-320: "Detecting a relationship between model bias and a climate variable using the RAT does not allow to directly conclude on a lack of model robustness." Isn't this statement in direct contradiction with your research hypothesis? Thereby, highlighting that the proposed RAT is not mature enough as a method for model robustness evaluation. From these contradictions, is it still relevant to use the RAT?

We believe that it is relevant, because it can help us compare models, it can help us identify unwanted/unexpected behaviors. RAT cannot reveal all the flaws of models but some: this is already a valid information useful for model selection.

L57: "...considered by all hydrologists as a good modelling practice". This statement is speculative. Something like "...most hydrologists..." would be acceptable.

Ok.

L74: I found the section 1.2 a bit too long.

We will consider reducing this section when revising the manuscript.

L101: IAHS should be defined here at its first occurrence, instead of at line 106.

We agree, this will be done.

L127-128: "it is difficult to distinguish which cases of DSST failure are truly caused by model structural inadequacy". Do you think that the RAT can address that limitation of the DSST? This is not demonstrated in your manuscript.

We think that because of the exhaustive nature of the RAT (examining the results of a long simulation period), it is less sensitive to the "contingency on the chosen calibration method" mentioned by Fowler et al. (2018).

L148-149: "The specificity of the RAT is that it requires only one calibration (or one parameterization)". The use of the term "calibration" is confusing. Shouldn't you use the term "simulation" here as you did at lines 140-141?

This will be changed when revising the manuscript: "*The specificity of the RAT is that it requires only one simulation covering…*"

L149: "at least 30 years". In Fig.1 it's "> 20 years". Is it 20 or 30 years? Be coherent through the manuscript (check lines 184, 310).

The reviewer is right, we will modify the text.

L270: "It happens on only two catchments out of 21". What are those two catchments? Do they have any similarities (climate, elevation, etc.) that might explain this result? Same questions for catchments identified under the other dependency tests (see L281, L287, L293).

We will consider developing these analyses in the revised version of the manuscript.

L300: "robustness evaluation from all types of hydrological models". This is not explicitly demonstrated in the manuscript. Only the GR4J is used in your methodology.

Since the RAT only needs one simulation time series, we believe that it is applicable to any kind of model. We remind that this article should more be seen as the description of a method than as the real assessment of a model. For demonstrating a drawback of hydrological models, we found it more elegant to demonstrate it on our own model rather than to include other models, which explains why only GR4J is applied here.

L320-322: "Indeed…robustness". This statement needs clarifications.

We will clarify the statement.

**2.3 Conclusion**

In conclusion, I think the RAT is subject to many limitations that challenge its own robustness and validity, which might hinder its large adoption by the scientific community. Therefore, I recommend that the authors develop strategies to address most of the limitations and thoroughly test the robustness of the RAT before it can potentially be released in the public.

We thank the reviewer for their comments and useful remarks and we hope that our answers will help to convince them about the benefit of the RAT method and will help to address potential misunderstandings

**3. Technical corrections**

L53: Thirel et al., 2015. Please specify if its "a" or "b". Also check line 100.

OK

L153: "numeric criterion" ---> "numeric bias criterion"

OK

L163: one missing closing parenthesis ")".

The missing parenthesis is actually located at l. 164

L250: "computation price" ---> "computation cost"

OK

L325: "metrological" ---> "meteorological"

We do not agree: metrology is used here in the sense of linked to measurement.

L335: "it true" ---> "it is true"

We rather changed to "prove it right"

---

## Author Response (AR1)

**Technical Note – RAT: a Robustness Assessment Test for calibrated and uncalibrated hydrological models**

Pierre Nicolle[1,3], Vazken Andréassian[1,*], Paul Royer-Gaspard[1], Charles Perrin[1], Guillaume Thirel[1], Laurent Coron[2], Léonard Santos[1]

**Answer to the comments of the reviewers and the editor**

We answer here formally to the remarks of the reviewers and of the editor.

We answer straightly to the questions (the thanks were already expressed during the discussion).

The color code is the following:

- the review is in black.
- our answers are in blue,
- the modifications introduced in the paper are in red,

**Answers to reviewer 1**

**1. General comments**

This technical note by Nicolle et al. presents an evaluation method for hydrological model robustness and is called the Robustness Assessment Test (RAT). The main assumption of the RAT is that the bias in the model output (i.e. streamflow) should not be correlated with the climatic input data (e.g. precipitation, temperature, air humidity), in which case, the model has a dependency on climactic variables, thereby not suitable for use in climate change impact studies. The manuscript is relatively well structured and written. The development of a framework for hydrological model evaluation is relevant, and can potentially be of interest for the readers of HESS if the RAT is thoroughly evaluated and some of its limitations are addressed. The RAT seems to be developed to detect deficiencies in model structure but it is not clearly demonstrated that model rejection by the RAT is not also due to input data. This is a key point that the authors have to address. Please refer to my comments and suggestions below.

**2. Specific comments**

**2.1 Major Comments**

**2.1.1 Contradictions**

The main assumption is that there should not be a dependency between model output bias and the input climatic variable for a hydrological model to be considered robust (Lines 152-157). However, in the conclusion, the authors clearly recognize that "Detecting a relationship between model bias and a climate variable using the RAT does not allow to directly conclude on a lack of model robustness". They further mention in the key points that "success at the RAT test is a necessary (but not sufficient) condition of model robustness". Therefore, I wonder why the RAT should be used for model evaluation.

We consider that under the assumption that the input data are reliable (which is to be checked before any study), a fail in passing the RAT is very likely indicating a model robustness issue. We added the following sections (starting l. 300)

**4.3     How to use RAT results?**

A question that many modelers may ask us is what can be done when different types of model failure are identified? Some of the authors of this paper have long be fond of the concept of Crash test (Andréassian et al., 2009), and we would like to argue here that the RAT too can be seen as a kind of crash-test. As all crash tests, it will end up identifying failures. But the fact that a car may be destroyed when projected against a wall does not mean that it is entirely unsafe, it rather means that it is not entirely safe. Although we are conscious of this, we keep driving cars… but, we are also willing to pay (invest) more for a safer car (even if this safer-and-more-expensive toy did also ultimately fail the crash test). We believe that the same will occur with hydrological models: The RAT may help identify safer models, or safer ways to parameterize models. If applied on large datasets, it may help identify model flaws, and thus help us work to eliminate them. It will not however help identify perfect models: these do not exist.

It seems unavoidable that forcing data quality will impact the results of RAT, but we would argue that it would similarly have an impact on the results of a Differential Split Sample Test. We believe that there is no way to avoid entirely this dependency, and that evaluating the quality of input data should be done before looking at model robustness;

**2.1.2 Input data**

How do you know if the model failure at the RAT is due to the dependency on input data or to the dependency on model structure or parametrization?

We assume that the reviewer means here "dependency on input data bias", not solely "dependency on input data". As mentioned below, we consider that checking the quality of input data has to be done prior to assessing the robustness of models. The RAT does not allow deciphering the sources of these dependencies. The same problem exists with the original split-sample test.

L170-171: Is the model rejected because the parametrization is wrong or because the input data is wrong? Would you get the same results with a different precipitation data? A clear answer to this question is essential for this work, as we do not want to reject a model that is not wrong.

In their work, the authors assume that in case there is a correlation between the model output bias and the input data, that correlation is due to the model structure/parametrization, as the authors do not investigate the potential contribution of the input data to the model output bias. This is a clear limitation of the evaluation of the RAT method. Without testing different input data, the authors implicitly assume that the only input data that they are using is right or at least is not the source of errors in the model outputs. But we know that input data remain a main source of uncertainty in hydrological modelling (Gupta, 1998 https://agupubs.onlinelibrary.wiley.com/doi/abs/10.1029/97WR03495). The authors recognize in the conclusion that the lack of model robustness can also be due to meteorological causes (L324-325). I urge the authors to test different input datasets, which are largely available nowadays because of the availability of satellite and reanalysis products. Different precipitation and temperature datasets must be tested to demonstrate that model rejection based on the RAT is not false negative, i.e. Type 2 error that we want to avoid in model evaluation (Beven, 2010 https://onlinelibrary.wiley.com/doi/epdf/10.1002/hyp.7718).

Another option to test if the model performance depends on precipitation, for instance, is to take a precipitation product and gradually apply some perturbation of [-30, -20, -10, 0, 10, 20, 30] percent bias and check if the bias in streamflow is correlated to precipitation for the different scenarios.

Following the suggestion of reviewer 1, we applied RAT with two different precipitation products. We used the CAMELS data set in the USA[1] :

*Table 1 : characteristics of the CAMELS dataset*

| Number of catchments | 673 |
|---|---|
| forcing 1 | Daymet, daily 1km grid derived solely from temperature and precipitation observations extrapolated through geostatistics |
* * *
[1] Addor, N., Newman, A. J., Mizukami, N., and Clark, M. P.: The CAMELS data set: catchment attributes and meteorology for large-sample studies, Hydrol. Earth Syst. Sci., 21, 5293–5313, https://doi.org/10.5194/hess-21-5293-2017, 2017.

| | dependent of local station) with quality control |
|---|---|
| forcing 2 | NLDAS (National Land Data Assimilation System) 12 km grid product based on North American Regional Reanalysis upscaled using 4 land surface models and adjusted using CPC for precipitations. |
| period | 1980 to 2014 (5 years for warm-up, calibration between 1985 and 2014) |

The model used was the same as in our paper (GR4J + Cemaneige snow accounting routine), we used the KGE on the square root of the discharge as objective function and the Oudin formula for PET. RAT was applied on the simulated time series 1985-2014 for both forcing data product. Results are shown in Table 2

*Table 2 : number of catchments considered reactive by RAT*

| Meteorological product | Average KGE of GR4J | Median KGE of GR4J | Number of catchments that react to RAT | |
|---|---|---|---|---|
| | | | (predictor: temperature) | (predictor: precipitation) |
| Daymet | 0.678 | 0.775 | 92 | 189 |
| NLDAS | 0.641 | 0.739 | 117 | 222 |
| Number of catchments which react with both products | | | 25 | 123 |

Model performance was better for Daymet (median KGE = 0.775 instead of 0.739; and mean KGE = 0.678 instead of 0.641). Obviously, the type of forcing used does have an impact on RAT results. It is interesting to note that the climatic dataset yielding the best simulation results is also the dataset yielding the less "reactive" catchments (22 % less for temperature and 15% less for precipitation).

It seems unavoidable that forcing data quality will impact the results of RAT since it has a huge impact on model simulation. It would similarly have an impact on the results of a Differential Split Sample Test. We would argue that there is no way to avoid entirely this dependency, evaluating the quality of input data should be done before looking at model robustness. To conclude, this dependency to data quality is not a sufficient reason to say that RAT is useless, even if it is a sufficient reason… to encourage modelers to take the test results with care and complement the RAT analysis with a discussion of data quality.

This was reflected in the conclusion as follows (l. 362)

7. Upon recommendation by one of the reviewers, we tried to assess the possible impact of the quality of the precipitation forcing on RAT results (see https://doi.org/10.5194/hess-2021-147-AC5) and found that the type of forcing used does have an impact on RAT results (interestingly, the climatic dataset yielding the best simulation results was also the dataset yielding the less catchments failing the robustness test). It seems unavoidable that forcing data quality will impact the results of RAT, but we would argue that it would similarly have an impact on the results of a Differential Split Sample Test. We believe that there is no way to avoid entirely this dependency, and that evaluating the quality of input data should be done before looking at model robustness;

**2.1.3 Seasonality**

Is the RAT valid for catchments with a strong seasonality in climatic variables? In case of bias in the input data, wouldn't that easily be reflected in the model output for catchments with strong seasonality? Thereby, misleading the conclusion of the RAT as the RAT would reject the model assuming that it is its parametrization that is problematic?

We believe that the RAT is still valid for catchments with a strong seasonality. However, we recognize that seasonal biases, if they do compensate each other, would not be detected by the RAT in its present form. A seasonal bias could indeed help.

**2.1.4 Annual aggregation - Hydrological year**

L159: Fig.1 - What is the reasoning behind the annual aggregation of the data by hydrological year to compute the bias? What are the implications of the choice of the hydrological year on the calculation of the bias?

You are right, a good model does not need a specific season to compute bias. But in the case of snow-affected catchments, it seems always more careful to avoid separating the snow accumulation and the snow melt seasons. We did test the twelve possible "hydrological years", and found that it only marginally affects the result of the test (see below).

[Figure]

*Figure 1*

We added the following in line 358:.

6.       Some of the modalities of the RAT, that we initially thought of importance, are not really important: this is for example the case with the use of hydrological years. We tested the twelve possible annual aggregations schemes (see https://doi.org/10.5194/hess-2021-147-AC6) and found no significant impact;

The model output bias and climatic variables may not be dependent at daily scale but show dependency at coarser temporal scale, or vice versa. How would that be captured by the RAT? The RAT might reject

the model at annual scale while there is no dependency at daily scale (temporal resolution of the hydrological simulations), or inversely.

We consider that because hydrological models are full of imperfections, it is more reasonable to work on bias at coarse time steps first, and then only to move to fine time steps. But we agree that the absence of bias for any given year, may well hide a positive bias during half of the year and a negative year during the other half. This is why we underline in conclusion that "the RAT should not be seen as the only basis for model choice."

**2.2 Minor comments**

L15: I see the RAT as a "complement" rather than an alternative to the "split-sample test".

We disagree on this comment. Indeed, the split-sample test cannot be applied to all models, e.g. physically-based models that do not use calibration. For these models, the RAT is therefore an alternative, not a complement.

A model might pass an SST but fail the RAT, or vice versa. What would happen in those cases? Should the model be rejected or accepted? Which of the SST and RAT outcomes would you give a preference?

This is a tough question, and it also depends on the threshold chosen (the p-value in the case of the RAT, the drop in criterion value between the calibration and the validation period in the case of the SST). Usually, people trust more a method that they have been using for a long time (experience with a method is an important factor of adoption). We have a little more than one year of experience with RAT, much more with SST... However we like the visualization opportunity that the RAT offers, this is why we are tempted to answer that we would give our preference to the RAT results.

L16: "the RAT method does not require multiple calibrations". This sentence can be misinterpreted because it seems like the RAT method could be calibrated while you are referring to the hydrological model. Should be "...multiple calibrations of hydrological models". Also correct this at line 137.

We agree:

l.16 now reads the RAT method does not require multiple calibrations of hydrological models: it is therefore applicable to uncalibrated models

l. 138 now reads : One of the problems of existing methods is the requirement of multiple calibrations of hydrological models

L20: As you said that success at the RAT is a necessary BUT not a sufficient condition of model robustness, can you call your approach a "robustness" assessment test? if a model is robust it would be successful at the RAT, but success at the RAT does not necessarily mean the model is robust. This key point highlights a strong limitation of RAT because you cannot confirm the robustness of the model but just have a hint that it might be robust.

We don't know of perfect tests (there is always a type I and type II error possibility, as you mentioned it). RAT is definitely not a "Robustness Assessment Certificate", but we do think that it is fair to call it a test.

L319-320: "Detecting a relationship between model bias and a climate variable using the RAT does not allow to directly conclude on a lack of model robustness." Isn't this statement in direct contradiction with

your research hypothesis? Thereby, highlighting that the proposed RAT is not mature enough as a method for model robustness evaluation. From these contradictions, is it still relevant to use the RAT?

We believe that it is relevant, because it can help us compare models, it can help us identify unwanted/unexpected behaviors. RAT cannot reveal all the flaws of models but some: this is already a valid information useful for model selection.

L57: "…considered by all hydrologists as a good modelling practice". This statement is speculative. Something like "…most hydrologists…" would be acceptable.

Agree : most hydrologists

L74: I found the section 1.2 a bit too long.

We removed a few sentences to reduce its length (see l.78, 92, 114)

L101: IAHS should be defined here at its first occurrence, instead of at line 106.

Done

L127-128: "it is difficult to distinguish which cases of DSST failure are truly caused by model structural inadequacy". Do you think that the RAT can address that limitation of the DSST? This is not demonstrated in your manuscript.

We think that because of the exhaustive nature of the RAT (examining the results of a long simulation period), it is less sensitive to the "contingency on the chosen calibration method" mentioned by Fowler et al. (2018).

L148-149: "The specificity of the RAT is that it requires only one calibration (or one parameterization)". The use of the term "calibration" is confusing. Shouldn't you use the term "simulation" here as you did at lines 140-141?

Done: The specificity of the RAT is that it requires only one simulation

L149: "at least 30 years". In Fig.1 it's "> 20 years". Is it 20 or 30 years? Be coherent through the manuscript (check lines 184, 310).

Done

L270: "It happens on only two catchments out of 21". What are those two catchments? Do they have any similarities (climate, elevation, etc.) that might explain this result? Same questions for catchments identified under the other dependency tests (see L281, L287, L293).

We have no explanation at this point, and in fact we cannot make statistics on 21 values. Only a test on a much larger catchment set could help in this regard. But our manuscript would not be a technical note anymore, but rather a very very long paper. We do prefer keeping this note clearly methodological and rather simple, and keep the large sample interpretation for a future paper. This is not "salami publishing": the fact is that there is already many details in this note to be explained and understood by the reader.

L300: "robustness evaluation from all types of hydrological models". This is not explicitly demonstrated in the manuscript. Only the GR4J is used in your methodology.

Since the RAT only needs one simulation time series, it is applicable to any kind of model. We remind that this technical note should more be seen as the description of a method than as the real assessment of a model. For demonstrating a drawback of hydrological models, we found it more elegant to demonstrate it on our own model rather than to include other models, which explains why only GR4J is applied here.

L320-322: "Indeed…robustness". This statement needs clarifications.

The sentence was rewritten as follows:

Detecting a relationship between model bias and a climate variable using the RAT does not allow to directly conclude on a lack of model robustness, because even a robust model will be affected by a trend in input data, yielding the impression that the hydrological model lacks robustness.

**2.3 Conclusion**

In conclusion, I think the RAT is subject to many limitations that challenge its own robustness and validity, which might hinder its large adoption by the scientific community. Therefore, I recommend that the authors develop strategies to address most of the limitations and thoroughly test the robustness of the RAT before it can potentially be released in the public.

We thank you for your comments and useful remarks and we hope that our answers will help to convince you of the possible benefit of the RAT method and will help to address potential misunderstandings

**3. Technical corrections**

L53: Thirel et al., 2015. Please specify if its "a" or "b". Also check line 100.

Done

L153: "numeric criterion" ---> "numeric bias criterion"

Done

L163: one missing closing parenthesis ")".

The missing parenthesis is actually located at l. 164

L250: "computation price" ---> "computation cost"

Done

L325: "metrological" ---> "meteorological"

Disagree : metrology is used here in the sense of linked to measurement.

L335: "it true" ---> "it is true"

We rather changed to "prove it right"

**Answers to reviewer 2**

This manuscript introduces an alternative to the generalised split sample test, which is less demanding computationally yet still provides similar insights into model robustness, as illustrated by Figures 4 and 5. This is an important outcome, as this new approach, named the RAT, has the potential to make the crash-testing of hydrological models more widely used before they are employed to assess future climate change impacts. Of course, more detailed tests of model realism exist, but there is, I believe, a need for tests that can be readily applied using typically available simulation data and provide a first-order assessment of the robustness of a model in a changing climate.

Thank you.

There is some ambiguity in the paper about what can and cannot be inferred from the RAT. This is for instance clear from the key point 3 ("the RAT method can be used to determine whether a hydrological model can be safely used for climate change impact studies") and 4 ("success at the RAT test is a necessary (but not sufficient) condition of model robustness"), which in my view, are in contradiction. While I agree with key point 4, I would argue that the RAT does not enable us to declare that it is "safe" to use the model for climate studies (at most, one could argue that the RAT is useful to identify models that are "unsafe" to use). Like the vast majority of model evaluation techniques, RAT can only falsify a model but not guarantee its validity. This is recognised by the authors on L322 L335 and in several other places, but it needs to be clearer throughout the key points, abstract and paper. Also, as the authors use the word ""climate‑proof"" L136, they should clarify that a methodology like the RAT is not enough to declare that a model is "climate-proof".

You are right, there is a logical contradiction with key point 4. Indeed, the RAT is useful to identify models that are "unsafe" to use. Or even (because we believe that no model can be considered as perfectly safe), RAT can be used to compare models and identify "the less unsafe" one. We modified this sentence as follows

the RAT method can be used to determine whether a hydrological model cannot be safely used for climate change impact studies

While I find section 4.1 quite convincing, I feel that section 4.2 needs more work. At the moment, it essentially illustrates that biases in streamflow simulations depend on different climatic variables for different catchments, which could be expected. There is scope for a more substantial discussion on what could be done when these different types of model failure happen. Should models be excluded as soon as streamflow errors are significantly correlated with one climate variable? Or two? Are correlations with some climatic variables more detrimental than others? Could the model be re-calibrated to improve robustness (if so, how?) or should other model structure(s) be used? Of course, one could simply say that "it depends on the study". But I think that answering these questions that users of the RAT are likely to face, or at least, proposing an approach to answer them, is essential for the RAT to be used widely, effectively, and in a consistent way across studies. I agree with the authors that "the RAT should not be seen as the only basis for model choice" L345

We would argue that we present here RAT as a tool. Once a tool is proved useful, different users may still be willing to use it in different ways.

We also added section 4.3:

**4.3 How to use RAT results?**

A question that many modelers may ask us is what can be done when different types of model failure are identified? Some of the authors of this paper have long be fond of the concept of Crash test (Andréassian et al., 2009), and we would like to argue here that the RAT too can be seen as a kind of crash-test. As all crash tests, it will end up identifying failures. But the fact that a car may be destroyed when projected against a wall does not mean that it is entirely unsafe, it rather means that it is not entirely safe. Although we are conscious of this, we keep driving cars… but, we are also willing to pay (invest) more for a safer car (even if this safer-and-more-expensive toy did also ultimately fail the crash test). We believe that the same will occur with hydrological models: The RAT may help identify safer models, or safer ways to parameterize models. If applied on large datasets, it may help identify model flaws, and thus help us work to eliminate them. It will not however help identify perfect models: these do not exist.

As the RAT is mostly data driven (in contrast to other tests focussed more on process representation), clearer recommendations on the input data should be provided. It is mentioned, almost in passing, that "some of the lacks of robustness detected among the 21 catchments presented here could be in fact due to metrological causes" (second bullet point of the conclusions). Hence, robust models can (wrongly) be rejected because of artefacts in the input data. Could the authors illustrate this, ideally using data from one of their catchments? Sadly, in large-sample datasets (often in contrast to studies relying on a few research catchments), there is rarely detailed enough metadata/knowledge on individual catchments to catch these artefacts in the input data. Furthermore, many large-sample datasets rely on meteorological gridded data products to produce a catchment average, and these products typically favour accuracy (using data from all available stations at each time step) over temporal homogeneity (sticking to the same set of stations for the entire period), so the risk of inhomogeneities is real. There may also be inhomogeneities in the streamflow time series, for instance caused by changes in rating. This is of course not something I am expecting the authors to solve, but I encourage them to discuss these data-related challenges earlier than in the conclusions (maybe in section 2 or 3). Ideally, we would like to differentiate between failures of RAT caused by the lack of robustness of the hydrological model (model inadequacy) and failures caused by trends/inhomogeneities in the data. Can the authors elaborate on this?

Honestly, we see more perspectives for learning from RAT on large sample experiments than on a catchment-by-catchment application. RAT may not offer a definitive answer but it can be used for comparing modelling alternatives.

Point 5 of the conclusions: "Although it would be tempting to transform the RAT into a post-processing method, we do not recommend it", what do the authors mean by a post-processing method? "What we do recommend is to work as much as possible on the model structure, to turn it less climate dependent", yes, but where to start?

A last precision on our use of the term "post-processing": if we identify a linear dependency between model bias and temperature for example, one could be tempted to fit a linear correction model in order to unbias the results (this is what we call "post-processing"). And when writing that we do not recommend it, we wanted to stress that we should not focus on the symptoms but rather try to identify the causes of the bias.

Thank you for developing this method.

**Answers to the editor**

Dear Editor,

Thank you for taking care of our Technical Note, and for your comments. Please find below short answers to your questions.

This technical note proposes an additional method to check whether it is reasonable to use a model under climate change conditions, a method that does not rely on numerous re-calibration. I see the value of the idea presented here but in addition to the reviewers' comments, I would like to raise some additional points.

First of all, the choice of the hydrological year might in my view have a considerable impact on the results and I am surprised that it is not tested in this note (easy to do: check the robustness of the results for other closing dates). Why? The hydrological year is most probably chosen such as to have little carry over of water storage from one year to the next (something that might need to be specified in the paper); for soil moisture and snow, August (as chosen in the paper) is certainly a reasonable choice but what about catchments that show the driest month in September? Will the carry-over effect result in spurious model bias correlations to temperature?

We tested the twelve possible "hydrological years", and found that it only marginally affects the result of the test (see below).

[Figure]

We added the following sentence to the conclusion:

6.        Some of the modalities of the RAT, that we initially thought of importance, are not really important: this is for example the case with the use of hydrological years. We tested the twelve possible

annual aggregations schemes (see https://doi.org/10.5194/hess-2021-147-AC6) and found no significant impact;

This brings me to the next point: the authors use a simple model, which has the main advantage that it should be straight-forward to explain what model "problem" could cause a correlation between bias and air temperature (or precipitation). It would be nice to see how we could attempt to interpret the climate dependency of the model. What model parameterizations can in fact influence the annual bias in this model? What does it tell us about the model if the bias increases with temperature or with precipitation? Obviously, such an interpretation is not possible for complex models but I really see this as a missed opportunity to share the authors' expertise in the use of a conceptual model to give guidance on how to further interpret the results. I.e. I would like to see a further elaboration of Section 4.2 (in addition to the comments raised by reviewer 2 on this section).

We do agree that it is very interesting to use a model (or several models…) on a wider catchment set to identify statistically the deficiencies of the model structure, which may explain their non-robustness. But if we introduce such a section in our note… we will need to describe the dataset, the model(s), interpret the results, propose a typology of "reacting" catchments… this will not be a technical note anymore, but rather a very very long paper. We would prefer to keep this note short and rather simple, and keep the large sample interpretation for a future paper. We do not like "Salami publishing", but the fact is that there is already many details to be explained and understood.

This attempt for interpretation might in fact even unravel unexpected reasoning: is it not a good sign if e.g. a simple snow model gives stronger bias in snow rich years as opposed to snow poor years?

This is exact.

Furthermore: I do not understand how the GSST points in Figure 4 are obtained; how is the bias for two simulations over two different periods (validation, calibration) defined.

The black points in Figure 4 correspond the GSST as initially published by Coron et al. (2012): all possible 10-year periods are used to calibrate the model and for each calibration, each 10-year sliding period over the remaining available period, strictly independent of the calibration one, is used to evaluate the model. The red points correspond to the "RAT" approach with a single simulation (and in our case, calibration) and as many validation periods as there are years.

And there is a formulation error in line 223, which reads like: "We then identified the catchments where the RAT procedure detected a lack of dependency of streamflow bias to climate variables." I guess it should be "identified a dependency"

No, our sentence was right, but we agree that it was uselessly complicated. We changed it:

We then identified the catchments where the RAT procedure detected a dependency of streamflow bias to one or several climate variables.

**References**

Coron, L., Andréassian, V., Perrin, C., Lerat, J., Vaze, J., Bourqui, M., & Hendrickx, F. (2012). Crash testing hydrological models in contrasted climate conditions: An experiment on 216 Australian catchments. Water Resources Research, 48, W05552. https://doi.org/10.1029/2011WR011721